# Δ Attention: Fast and Accurate Sparse Attention Inference by Delta Correction

**Jeffrey Willette**[1], **Heejun Lee**[1], **Sung Ju Hwang**[1,2]

KAIST[1], DeepAuto.ai[2]

{jwillette, ainl, sjhwang82}@kaist.ac.kr

## Abstract

The attention mechanism of a transformer has a quadratic complexity, leading to high inference costs and latency for long sequences. However, attention matrices are mostly sparse, which implies that many entries may be omitted from computation for efficient inference. Sparse attention inference methods aim to reduce this computational burden; however, they also come with a troublesome performance degradation. We discover that one reason for this degradation is that the sparse calculation induces a distributional shift in the attention outputs. The distributional shift causes decoding-time queries to fail to align well with the appropriate keys from the prefill stage, leading to a drop in performance. We propose a simple, novel, and effective procedure for correcting this distributional shift, bringing the distribution of sparse attention outputs closer to that of quadratic attention. Our method can be applied on top of any sparse attention method, and results in an average 36%pt performance increase, recovering 88% of quadratic attention accuracy on the 131K RULER benchmark when applied on top of sliding window attention with sink tokens while only adding a small overhead. Our method can maintain approximately 98.5% sparsity over full quadratic attention, making our model 32 times faster than Flash Attention 2 when processing 1M token prefills.

## 1 Introduction

The main operation that powers modern transformers, self-attention [Vaswani et al., 2017], creates causal pairwise comparisons for every item in a sequence. While powerful and expressive, this operation comes with a quadratic complexity, leading to the need for large amounts of computation during inference on long sequences. This increases direct costs for hardware and electricity as well as negative externalities such as $CO_2$ emissions. Training-free sparse attention modifications aim to lower the quadratic complexity at inference time, but come with unwanted side effects such as accuracy degradation due to the sparsification of the attention matrix.

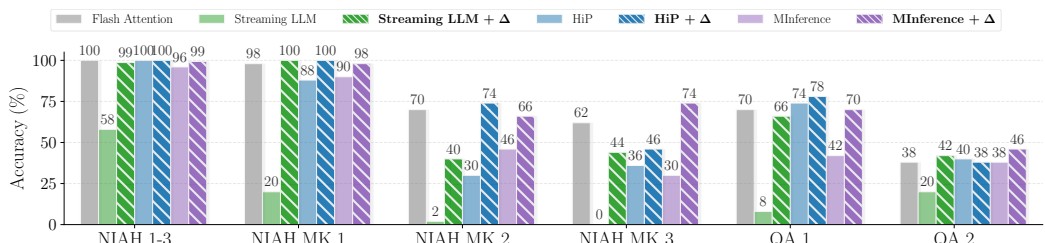

Figure 1: **RULER 131K Subsets.** At long context lengths, sparse attention can degrade performance by a large margin. Our simple **Δ** correction improves performance and only requires an additional 1.5% of the full quadratic attention computation.

39th Conference on Neural Information Processing Systems (NeurIPS 2025).

Recent works on sparse attention have found that a sparse sliding window can be added at inference time without a total loss of model stability. This is accomplished by saving a small number of initial tokens, and applying a sliding window on all subsequent tokens (Streaming LLM [Xiao et al., 2023]). Subsequent works such as Star Attention [Acharya et al., 2024] have proposed a similar sparse prefill strategy with a fully dense decoding procedure to generate new tokens. This strategy has the positive attribute of a sparse prefill while still performing attention with all tokens during generation. This should allow the model to accurately recall context buried deep within the prompt. However, we find that this is not the case in practice. For example, there is a challenging subset of the RULER [Hsieh et al., 2024] benchmark titled MultiKey-3, which consists entirely of unique UUID keys and values, and the large language model (LLM) must be able to recall the proper value for a particular key in order to get a correct answer. In this setting, a sliding window of 2048 tokens provides more than adequate room for encoding individual key and value pairs together within the window. One would then expect that a dense decode procedure would be able to retrieve the proper UUID given a user query. However, we find that this is not the case and the dense decode achieves a surprisingly low accuracy of 0% as opposed to 62% when using quadratic attention.

We find this drop in accuracy arises from a distributional shift in the output tokens of each layer due to the sparse prefill. This distributional shift causes problems with the query-key dot products in long contexts and therefore results in an extreme drop in performance as the queries no longer align with the expected keys. We study this problem and found a surprisingly simple fix which we dub $\mathbf{\Delta}$ Attention that improves the accuracy of sliding window attention from **0% to 44% (Figure 1, NIAH MK3) on this challenging subset** while maintaining more than 11-fold speedup over plain Flash Attention 2 [Dao, 2023] for processing 131K context lengths (Figure 2). Through evaluations on perplexity, natural language understanding, and synthetic tasks, we demonstrate that our method consistently results in better performance while maintaining the low latency of the sparse prefill.

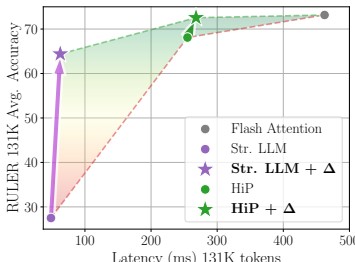

Figure 2: Comparing RULER 131K prefill attention latency and accuracy for sparse attention methods.

Our contributions are as follows:

- We identify a distributional shift in tokens when applying an inference-time sparse attention method to pretrained transformers, which interferes with query-key alignment on long contexts and leads to a drop in performance.

- We introduce Delta ($\mathbf{\Delta}$) Attention, a sparse post-processing correction that realigns sparse outputs with full quadratic attention.

- Our method adds negligible latency overhead compared to plain sparse attention, while drastically increasing performance over purely sparse methods.

- Our method is designed to work in the attention output space, so it can be seamlessly integrated with existing sparse attention kernels and inference pipelines without major modification.

## 2  Background & Related Work

The self attention mechanism of a transformer takes an input sequence $\mathbf{X} \in \mathbb{R}^{N \times d}$ of individual tokens $\mathbf{x}_i \in \mathbb{R}^d$ for $i \in \{1..N\}$. After applying linear projections $\mathbf{W_Q}, \mathbf{W_K}, \mathbf{W_V} \in \mathbb{R}^{d \times d}$ to the input $\mathbf{X}$ to achieve the respective $\mathbf{Q}, \mathbf{K}, \mathbf{V}$ matrices, positional encodings such as [Su et al., 2024] are applied to $\mathbf{Q}$ and $\mathbf{K}$. With $\sigma$ representing the softmax operation over the last dimension, the self-attention operation for an arbitrary layer in a transformer is the following,

$$\mathbf{AV} = \sigma\left(\frac{\mathbf{QK}^\top}{\sqrt{d}}\right)\mathbf{V} = \sigma\left(\frac{\mathbf{XW_Q}(\mathbf{XW_K})^\top}{\sqrt{d}}\right)\mathbf{XW_V} \qquad (1)$$

We omit the output projections, attention heads, and post-attention multilayer perceptrons (MLPs). For a deeper discussion of these topics in transformers, please see [Vaswani et al., 2017]. The most expensive operation in Equation (1) that arises from the multiplication inside $\sigma()$ which results in the implicit construction of an attention matrix $\mathbf{A} \in \mathbb{R}^{N \times N}$ which is computationally expensive for

large $N$. Due to the causality condition of language, a token $\mathbf{x}_i$ may only influence another token $\mathbf{x}_j$ where the index $i \leq j$. In practice, this means that only the lower triangle of $\mathbf{A}$ is computed.

After traversing through the layers of the network, the next token in the sequence $\mathbf{x}_{N+1}$ is generated (predicted) and added to the input sequence to generate the next token and so on until the sequence terminates. In this generation phase, each iteration may use the previously computed tokens, which are stored within a cache at each layer, so that we may avoid re-calculating the entire attention matrix in Equation (1). With a union operator $\cup$ which concatenates matrices by adding new rows, and considering that $\mathbf{K}, \mathbf{V}$ contain tokens with indices $\{1..N\}$, and the newly generated token has index $i = N + 1$, the generative process for the next token proceeds through the attention layers as,

$$(\mathbf{a}^\top \mathbf{v})_i = \sigma \left( \frac{\mathbf{q}_i^\top \left[ \mathbf{K} \cup \mathbf{k}_i^\top \right]^\top}{\sqrt{d}} \right) \left( \mathbf{V} \cup \mathbf{v}_i^\top \right) \tag{2}$$

Sparse attention prefill methods aim to reduce the quadratic computation in Equation (1) by computing a subset of entries within $\mathbf{A}$, forming a sparse matrix $\mathbf{A}^*$ where the number of computed entries $\sum_{i,j} \mathbb{1}\{\mathbf{A}^*_{i,j} > 0\} \ll \frac{N^2}{2}$ with minimal information loss. However, in practice, large portions of the attention matrix are ignored, which may cause unintended differences in the output tokens and lead to unexpected behavior of future query-key dot products, which could degrade performance on downstream tasks. Previous works have studied in-context learning (ICL) processes such as induction heads [Olsson et al., 2022], which are responsible for copying relevant content from earlier tokens into later tokens in the sequence [Musat, 2024]. Induction heads are known to be more prevalent in the lower layers of the network [Yin and Steinhardt, 2025], which implies that a distributional mismatch between queries and keys at the lower layers of the network will inhibit ICL processes. Additionally, Wu et al. [2024] showed that these induction or retrieval heads are universal for all transformer model types and further highlighted that interfering with these special attention heads causes a catastrophic drop in performance on downstream tasks during inference.

Recent works on sparse attention, such as Streaming LLM [Xiao et al., 2023], have shown that a pretrained quadratic transformers can be modified on-the-fly at test time into a stable sparse attention model by utilizing sink tokens and sliding windows. This has inspired a multitude of recent works that utilize this knowledge for inference time adaptations that selectively prune the less important 'middle' tokens from the KV-cache during inference. Two approaches, H2O [Zhang et al., 2024b] and SnapKV [Li et al., 2024] accomplish this by looking at historical attention scores to decide which tokens to prune. However, these works still leave the quadratic prompt in place, which requires a computation overhead of $\mathcal{O}(n^2)$.

Other recent works have therefore made efforts to lower the complexity of the prompt as well. Big Bird [Zaheer et al., 2020] studies the effect of randomly choosing keys for every new query in the attention matrix. However, random key selection has been shown to underperform a more targeted selection of keys in HiP Attention [Lee et al., 2024a,b], which applies a tree-based pruning mechanism that masks out less important blocks of keys in order to sparsify the computation of the attention matrix. MInference [Jiang et al., 2024] studies reliably recurring patterns in the attention matrix of specific attention heads, and builds a set of sparse kernels which apply sparse attention following these patterns. Star Attention [Acharya et al., 2024] uses a sparse strategy akin to that of Streaming LLM with a sliding window, initial tokens, and a fully dense decode procedure which evaluates the dot product between every past key for new queries during the decoding phase. As we show in our experiments, this scheme does not work for all tasks unless the sliding window represents a large percentage of the total context length (see Table 1).

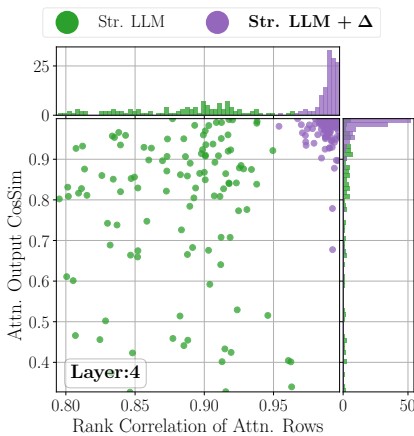

Figure 3: Comparing sparse attention methods to quadratic attention. Our $\Delta$ correction results in outputs that are more similar to quadratic attention.

To illustrate how our findings integrate with these prior works, we provide an example in Figure 3. In this experiment, we use quadratic attention and Streaming LLM to prefill a 131K length input from the RULER benchmark. We then compute the cosine similarity $\cos([\mathbf{A}^*\mathbf{V}]_i, [\mathbf{A}\mathbf{V}]_i)$ of the sparse and quadratic outputs, and also construct the last part of the full attention matrix using the

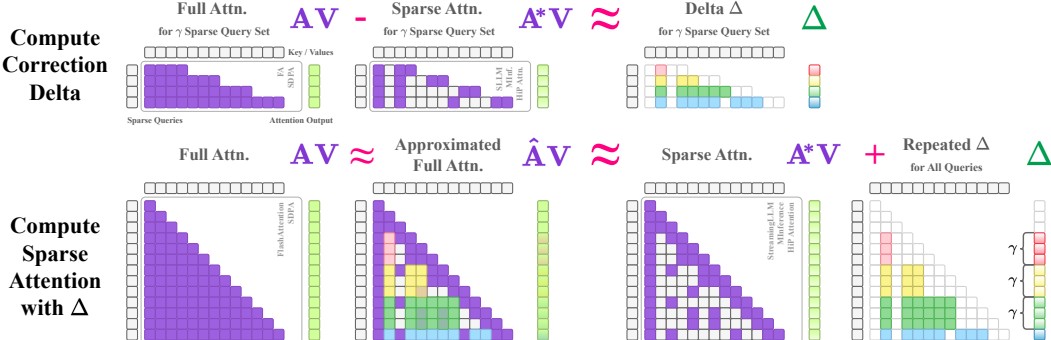

Figure 4: **Overview of $\mathbf{\Delta}$ Attention. (Top)** Given an arbitrary sparse attention method we calculate the difference between the sparse attention and full attention for a small subset of queries. The subset size is controlled by a hyperparameter $\gamma$. **(Bottom)** We then repeat the calculated difference for all output tokens and add the result to the full sparse attention output. The result is an approximation to the original quadratic attention.

last 128 queries in order to compare the rank correlation coefficient $\rho(\mathbf{A}^*{}_i, \mathbf{A}_i)$ in the final rows of the attention matrix. If the sparse attention method does not cause a distributional shift, then the attention outputs should have a high cosine similarity to quadratic attention, and sorting the rows of the attention matrix should lead to the same sort order, which implies that the relative importance (ranking) between queries and keys has been maintained. As seen in Figure 3, in both dimensions, the sparse attention of Streaming LLM causes a drift in the distribution of tokens, which causes the degradation in task performance seen in Figure 1. However, we find we can correct this distributional shift with the addition of a $\mathbf{\Delta}$ term which we will describe in the following section.

## 3 Method

Given the distributional shift shown in Figure 3, our method answers the following question: *How may we shift the distribution of attention outputs such that they are closer to the representation which is expected during quadratic attention?* Specifically, we wish to add a term to the sparse attention output $\mathbf{A}^*\mathbf{V}$ such that we recover the attention contribution $\mathbf{A}^{\mathbf{\Delta}}\mathbf{V}$ from the places where sparse attention has given zero weight. This region is usually located somewhere inside the lower triangle of the attention

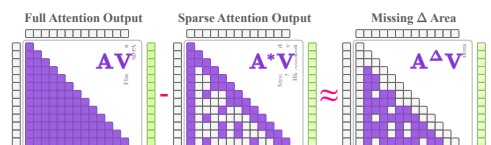

Figure 5: **Intuition for $\mathbf{\Delta}$ Attention.** The difference of attention outputs approximates the missing attention contribution.

matrix and resembles a delta shape. We propose to approximate this $\mathbf{\Delta}$ region by a simple difference of attention outputs, as geometrically depicted in Figure 5. Specifically,

$$\mathbf{A}^{\mathbf{\Delta}}\mathbf{V} \approx \mathbf{A}\mathbf{V} - \mathbf{A}^*\mathbf{V} \tag{3}$$

Note that the softmax normalization of sparse attention methods generally only computes the normalization constant over the nonzero values. Thus, $\mathbf{A}$ and $\mathbf{A}^*$ have different normalization constants, which makes the relation an approximation. We consider $\mathbf{A}$ and $\mathbf{A}^{\mathbf{\Delta}}$ to share the same softmax normalization constant. Let the full attention softmax normalization constant be $T + H$, and the sparse attention normalization constant be $T$.

**Lemma 1.** *w.l.o.g. Consider an arbitrary row in the attention matrix $\mathbf{a}$ and arbitrary column of the values $\mathbf{v}$, with both $\mathbf{a}$ and $\mathbf{v}$ being sorted according to rank of $\mathbf{a}$ such that $\mathbf{a} = (a_{r(1)} \leq a_{r(2)} \leq \cdots \leq a_{r(N)})$. For a top-$k$ sparse attention matrix which only computes the top-$k$ attention scores, one only needs to compute $\mathbf{a}^{*\top}\mathbf{v} = \sum_{N-k+1}^{N} \mathbf{a}^*{}_i\mathbf{v}_i$. With $\mathbf{\Delta} = \mathbf{a}^\top\mathbf{v} - \mathbf{a}^{*\top}\mathbf{v}$, we may bound the error of our attention approximation as,*

$$\left| \mathbf{\Delta} - \sum_{i=1}^{N-k} \mathbf{a}_i \, \mathbf{v}_i \right| \leq \frac{H}{H+T} \max_{i > N-k} |v_i|$$

*Proof.* See Section G. $\qquad\square$

We ultimately seek a shift in the attention outputs such that $\mathbf{A}^*\mathbf{V} + \mathbf{\Delta} \approx \mathbf{A}\mathbf{V}$. Trivially, if we choose $\mathbf{\Delta} = \mathbf{A}\mathbf{V} - \mathbf{A}^*\mathbf{V}$, we have exact equality; however, calculating $\mathbf{A}$ requires the full

quadratic attention procedure that we wish to avoid. As $\mathbf{AV} - \mathbf{A}^*\mathbf{V} \approx \mathbf{A}^{\boldsymbol{\Delta}}\mathbf{V}$, if we further assume that $(\mathbf{A}^{\boldsymbol{\Delta}}\mathbf{V})_i \approx (\mathbf{A}^{\boldsymbol{\Delta}}\mathbf{V})_{i+\nu}$ for $\nu \in \{1, \ldots, \gamma\}$ and $\gamma \in \mathbb{N}$, then we may approximate $(\mathbf{AV})_{i+\nu} \approx (\mathbf{A}^{\boldsymbol{\Delta}}\mathbf{V})_i + (\mathbf{A}^*\mathbf{V})_{i+\nu}$. Under this approximation, one only needs to compute every $\gamma^{\text{th}}$ row of the attention matrix, which maintains a sparse computation by only computing a subset of rows of $\mathbf{A}$. To do this, we select a fixed fraction of row indices from $\mathbf{Q}$, such that,

$$\widetilde{\mathbf{Q}}_{\lfloor \frac{i}{\gamma} \rfloor} = \mathbf{Q}_i \implies i \bmod \gamma = 0; \quad \forall \quad i \in \{1 \mathinner{..} N\} \tag{4}$$

and therefore $\widetilde{\mathbf{A}}\mathbf{V} = \sigma(\widetilde{\mathbf{Q}}\mathbf{K}^\top)\mathbf{V}$ which is sparse in the query dimension, but dense in the key dimension. One possible approach would be to substitute this representation into the appropriate rows of the sparse output $\mathbf{A}^*\mathbf{V}$ such that the final representation $\widehat{\mathbf{A}}$, is the following,

$$\left(\widehat{\mathbf{A}}\mathbf{V}\right)_i = (\mathbf{A}^*\mathbf{V})_i + \overbrace{\mathbb{1}\{i \bmod \gamma = 0\} \left[\widetilde{\mathbf{A}}\mathbf{V}_{\lfloor \frac{i}{\gamma} \rfloor} - (\mathbf{A}^*\mathbf{V})_{\lfloor \frac{i}{\gamma} \rfloor \gamma}\right]}^{\text{make a dense output row if } i \bmod \gamma = 0}; \quad \forall \quad i \in \{1 \mathinner{..} N\} \tag{5}$$

We dub this approach as 'recompute', as we are essentially using the sparse representation with some densely computed output tokens interwoven at regular intervals. However, we find that this approach still does not shift the distribution of attention outputs far enough towards the expected representation under quadratic attention (see Figure 9). Therefore, in order to apply a shift to all tokens in the output of $\mathbf{A}^*\mathbf{V}$ while maintaining a sparse computation, we instead apply the following correction to the sparse attention output,

$$\left(\widehat{\mathbf{A}}\mathbf{V}\right)_i = (\mathbf{A}^*\mathbf{V})_i + (\mathbf{A}^{\boldsymbol{\Delta}}\mathbf{V})_{\lfloor \frac{i}{\gamma} \rfloor \gamma} \tag{6}$$

$$= (\mathbf{A}^*\mathbf{V})_i + \underbrace{\left[\widetilde{\mathbf{A}}\mathbf{V}_{\lfloor \frac{i}{\gamma} \rfloor} - (\mathbf{A}^*\mathbf{V})_{\lfloor \frac{i}{\gamma} \rfloor \gamma}\right]}_{\boldsymbol{\Delta} \text{ correction term}} \tag{7}$$

Which is equivalent to swapping in a dense row of the attention matrix at every $\gamma^{\text{th}}$ row, and applying the difference between the dense and sparse attention for the previous $\gamma^{\text{th}}$ row otherwise. A visual depiction of this process can be seen in Figure 4, and pseudocode in Algorithm 1. Since our method is applied directly on the attention outputs, we may utilize existing sparse attention kernels to compute $\mathbf{A}^*\mathbf{V}$ and make use of a minimally modified flash attention kernel to compute our query-sparse attention $\widetilde{\mathbf{A}}\mathbf{V}$.

Assuming that a row index $j$ of the attention matrix is not evenly divisible by $\gamma$, this means that an attention differential from a previous row is being applied to the current row $j$. The intuition from this operation comes from prior works which have studied attention locality [Lee et al., 2024a], finding that the difference between attention scores for neighboring tokens is generally small. Likewise, our conjecture is that

---

**Algorithm 1: $\boldsymbol{\Delta}$ Attention Algorithm**

---

**Require:** $f(), f^*() \ \mathbf{Q}, \mathbf{K}, \mathbf{V}, \gamma$
  // sparse attention for all of $\mathbf{Q}$
  $\mathbf{A}^*\mathbf{V} \leftarrow f^*(\mathbf{Q}, \mathbf{K}, \mathbf{V})$
  $\widetilde{\mathbf{Q}} \leftarrow$ Equation 4
  // dense attention every $\gamma^{\text{th}}$ query
  $\widetilde{\mathbf{A}}\mathbf{V} \leftarrow f(\widetilde{\mathbf{Q}}, \mathbf{K}, \mathbf{V})$
  // collect proper indices for $\boldsymbol{\Delta}$ construction
  $\delta \leftarrow \{i \mid i \bmod \gamma = 0\}$
  $\boldsymbol{\Delta} \leftarrow \widetilde{\mathbf{A}}\mathbf{V} - (\mathbf{A}^*\mathbf{V})_{i \in \delta}$
  // repeat $\boldsymbol{\Delta}$ and apply correction
  $\widehat{\mathbf{A}}\mathbf{V} = \mathbf{A}^*\mathbf{V} + \text{repeat}(\boldsymbol{\Delta}, \gamma)$
  return $\widehat{\mathbf{A}}\mathbf{V}$

---

the low attention score regions from neighboring rows of the attention matrix also have a negligible difference, allowing for the less important part of the row of the attention matrix to be reused multiple times. Specifically, as stated above Equation (4), we assume that $(\mathbf{A}^{\boldsymbol{\Delta}}\mathbf{V})_i \approx (\mathbf{A}^{\boldsymbol{\Delta}}\mathbf{V})_{i+\nu}$ for $\nu \in \{1, \ldots, \gamma\}$ and $\gamma \in \mathbb{N}$. To validate this assumption, we examine the average cosine similarity of $(\mathbf{A}^{\boldsymbol{\Delta}}\mathbf{V})_i$ within a $\gamma$ window on an input from the RULER 131K task set for various values of $\gamma$ in Figure 6b. We find a high average cosine similarity within the window, implying that $(\mathbf{A}^{\boldsymbol{\Delta}}\mathbf{V})_i$ may be reused for multiple rows of the attention output.

## 4 Experiments

We evaluate our method in terms of perplexity (PPL) and long context perplexity using the LongPPL [Fang et al., 2024] metric on a QA version of the PG19 [Rae et al., 2019] test set, which was recently proposed as a long context understanding dataset [He et al., 2025]. We also provide

Table 1: **RULER (Llama 3.1 8B Instruct and Mistral NeMo 12B)** for sparse attention methods. Adding $\Delta$ Attention results in better overall accuracy, with the largest improvement occurring at the longest context length and on the most naive sparse method (Streaming LLM). Colors are relative to each attention method group + Flash Attention 2.

| Model | Llama 3.1 8B Instruct | | | | | | | | | | Mistral NeMo 12B | | | | |
|---|---|---|---|---|---|---|---|---|---|---|---|---|---|---|---|
| Attn. Method | Flash Attn. | Str. LLM | Str. LLM | Str. LLM | Str. LLM | **Str. LLM+Δ** | MInf. | **MInf.+Δ** | HiP | **HiP+Δ** | Flash Attn. | Str. LLM | **Str. LLM+Δ** | HiP | **HiP+Δ** |
| Wind. | - | 2K | 4K | 16K | 32K | 2K | 3K | 3K | 3K | 3K | - | 2K | 2k | 3K | 3K |
| 4K | 96.74 | 90.52 | 96.71 | 96.71 | 96.71 | 96.54 | 96.74 | 96.71 | 96.80 | 96.31 | 90.60 | 71.01 | 90.42 | 90.36 | 90.55 |
| 8K | 93.25 | 60.53 | 93.76 | 93.76 | 93.76 | 92.25 | 93.65 | 93.69 | 94.56 | 94.43 | 87.67 | 44.89 | 85.38 | 88.36 | 87.69 |
| 16K | 90.99 | 38.13 | 68.07 | 91.15 | 91.15 | 88.66 | 92.32 | 91.34 | 94.10 | 93.86 | 81.82 | 33.28 | 78.07 | 78.07 | 81.08 |
| 32K | 85.84 | 30.25 | 43.38 | 56.32 | 85.83 | 81.27 | 86.75 | 85.96 | 89.92 | 89.39 | 62.54 | 12.27 | 34.76 | 58.76 | 60.38 |
| 65K | 85.25 | 18.59 | 34.08 | 41.28 | 58.35 | 75.22 | 84.43 | 83.67 | 82.51 | 84.89 | 46.89 | 03.28 | 16.22 | 35.87 | 41.56 |
| 131K | 73.16 | 27.45 | 30.32 | 40.51 | 49.17 | 64.40 | 65.73 | 73.31 | 68.74 | 73.71 | 18.09 | 02.25 | 01.44 | 10.10 | 10.93 |
| Avg. | 87.54 | 44.25 | 61.05 | 69.96 | 79.16 | **83.06** | 86.60 | **87.44** | 87.77 | **88.76** | 64.60 | 27.83 | **51.05** | 60.25 | **62.03** |

evaluations of our method on the RULER [Hsieh et al., 2024] benchmark, which tests models' performance under a number of long context retrieval tasks. Additionally, we evaluate our $\Delta$ Attention on Infinite-Bench [Zhang et al., 2024a], and also provide analysis that evaluates the effect of our $\Delta$ correction on the distribution of attention outputs and scores, and overall attention latency. Our work considers that the decoding process shown in Equation (2) is dense along the key dimension and should be able to successfully learn from previously encoded information during the sparse prefill.

We apply our method in conjunction with the sparse attention methods Streaming LLM [Xiao et al., 2023], HiP [Lee et al., 2024a,b], and MInference [Jiang et al., 2024], on models from the Llama [Dubey et al., 2024] (3.1 and 4), and Mistral [Jiang et al., 2023] model families. Unless otherwise noted, our standard setting uses $\gamma = 64$ which means we calculate every 64th query row (approximately 98.5% sparsity) in the attention computation required by $\Delta$ Attention.

**RULER.** For baselines on needle-in-a-haystack type tasks, we compare our method in addition to Streaming LLM, HiP, and MInference for both Llama and Mistral models. In all cases, $\Delta$ Attention shows a large improvement upon the given sparse methods, and especially at the longer context lengths in Table 1. In particular, we note an improvement of nearly 37%pt over Streaming LLM with the same 2K window size for 131K with Llama 3.1. For Streaming LLM, if we adjust for the extra computation needed by our method, we find that the approximate window size of our method is 3072 (see Section F for calculation). This is due to the fact that we also use a sliding window of 2048 and compute every 64th row of the lower triangle in the attention matrix. Therefore, even when Streaming LLM is allowed a higher computational budget of a 4K window, $\Delta$ Attention still results in an increase of 34%pt, more than doubling the accuracy of Streaming LLM (+112%, relative). Even when Streaming LLM is allowed a 32K window, Streaming LLM + $\Delta$ with a 2K window still delivers higher accuracy.

**Perplexity (PPL) and Long Perplexity (LongPPL).** We generated a QA dataset based on the PG19 test set according to the procedure outlined by He et al. [2025]. This results in a long context task where an entire book is used as context, along with a series of LLM-generated questions and answer pairs with total context lengths of approximately 100K. In order to excel at this task, a model must be able to retain all information and facts from the text, which may be asked in the follow-up QA session. We

Table 2: **Perplexity on PG19 Long QA [He et al., 2025].** Our simple $\Delta$ correction results in a significant drop in both PPL and Long PPL.

| Method | Long PPL ↓ | PPL ↓ |
|---|---|---|
| Flash Attention 2 | 5.11 (-) | 3.33 (-) |
| Streaming LLM | 7.02 (+1.91) | 3.54 (+0.21) |
| **Streaming LLM + Δ** | **5.96 (+0.85)** | **3.41 (+0.08)** |
| HiP Attention | 6.29 (+1.18) | 3.48 (+0.15) |
| **HiP Attention + Δ** | **5.45 (+0.34)** | **3.37 (+0.04)** |

evaluate both PPL and LongPPL, where the latter metric selects a subset of tokens that are found to rely heavily on long context for the final loss calculation. LongPPL has been shown to have a stronger correlation with long context performance over PPL [Fang et al., 2024]. We use Llama 3.1 8B instruction-tuned models for this experiment. Results can be seen in Table 2 and Figure 6. When

Table 3: ∞-**bench results.** Colors are made relative to the best and worst metrics within each model group, with Flash Attention being part of every group. Our **Δ** correction improves overall performance in every case. En.QAR displays recall for the En.QA subset.

| Model | Method | Ctx Len. | En.MC | En.QA | En.QAR | En.Sum | Passkey | Number | KV | Math.F | Avg. |
|-------|--------|----------|-------|-------|--------|--------|---------|--------|----|--------|------|
| Llama 3.1 8B Instruct | Flash Attention | 126K | 64.19 | 35.89 | 44.69 | 31.59 | 99.13 | 99.83 | 92.40 | 24.86 | 61.57 |
| | HiP | 126K | 54.15 | 31.49 | 38.12 | 31.06 | 75.08 | 96.10 | 30.60 | 18.86 | 46.93 |
| | **HiP + Δ** | 126K | 61.14 | 33.70 | 43.54 | 31.30 | 100.0 | 97.97 | 69.60 | 25.71 | **57.87** |
| | Str. LLM | 126K | 27.95 | 07.25 | 14.67 | 20.57 | 02.71 | 01.36 | 01.20 | 25.14 | 12.51 |
| | **Str. LLM + Δ** | 126K | 56.33 | 24.93 | 33.35 | 26.95 | 96.27 | 68.81 | 00.40 | 25.43 | **41.66** |
| Llama 4 Scout 109B | Flash Attention | 384K | 82.10 | 44.34 | 48.82 | 35.30 | 100.0 | 100.0 | 99.20 | 43.14 | 69.11 |
| | HiP | 384K | 74.67 | 43.19 | 48.29 | 34.28 | 100.0 | 99.83 | 99.40 | 41.14 | 67.60 |
| | **HiP + Δ** | 384K | 78.60 | 42.84 | 48.14 | 34.06 | 100.0 | 99.66 | 97.20 | 44.29 | **68.10** |
| | Str. LLM | 384K | 49.78 | 15.23 | 26.11 | 31.50 | 52.88 | 08.31 | 03.40 | 40.57 | 28.47 |
| | **Str. LLM + Δ** | 384K | 73.80 | 37.82 | 43.03 | 30.62 | 94.75 | 91.36 | 46.60 | 40.86 | **57.35** |

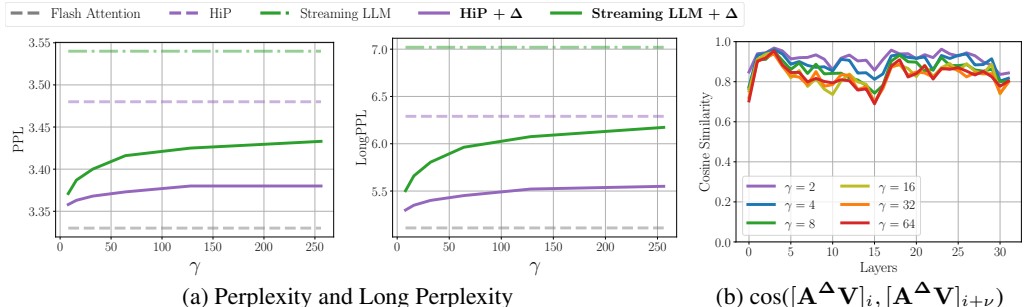

(a) Perplexity and Long Perplexity          (b) $\cos([\mathbf{A}^{\mathbf{\Delta}}\mathbf{V}]_i, [\mathbf{A}^{\mathbf{\Delta}}\mathbf{V}]_{i+\nu})$

Figure 6: (**a**) Perplexity metrics for increasing $\gamma \in \{2^3, \ldots, 2^8\}$ . For PPL and LongPPL, increasing the query stride shows a slight trend towards higher PPL with higher sparsity. (**b**) Measures the average cosine similarity between the approximate $(\mathbf{A}^{\mathbf{\Delta}}\mathbf{V})_i$ and $(\mathbf{A}^{\mathbf{\Delta}}\mathbf{V})_{i+\nu}$ for $\nu \in \{1, \ldots, \gamma\}$ for Streaming LLM and finds a high similarity within a $\gamma$ neighborhood of attention outputs. High similarity implies $(\mathbf{A}^{\mathbf{\Delta}}\mathbf{V})_i$ can be reused within the $\gamma$ neighborhood.

our **Δ** Attention is applied on top of both HiP and Streaming LLM, we achieve between a 50-75% reduction in the PPL performance gap between quadratic attention. This trend holds true for both PPL and LongPPL. Figure 6 shows the effect of varying the $\gamma$ parameter form 8-256. As $\gamma$ also controls the sparsity, we find that as the sparsity increases, both perplexity metrics tend to rise.

**Infinite Bench.** [Zhang et al., 2024a] For both LLama 3.1 8B and Llama 4 Scout 109B, results are displayed in Table 3. The display colors are encoded to show the performance difference within each model group, and including flash attention in all groups. For Llama 4 (Streaming LLM), the addition of **Δ** resulted in an increase of 40%pt, which leads to recapturing 82% of quadratic attention accuracy (up from 41%). Similarly, for Llama 3.1, the addition of **Δ** increased overall performance by 29%pt, which moves from 20% of full attention accuracy to recovering 67%. The realized performance gains when applying our method to HiP result in a 10%pt increase for Llama 3.1 and a 0.5%pt increase for Llama 4. Note that HiP with Llama 4 only shows a total of 1.5%pt gap in performance, which means that **Δ** Attention was able to recapture 33% of the total performance gap.

## 4.1 Ablation & Analysis

**Latency.** For a single attention layer, our method shows a large reduction in latency when compared to Flash Attention 2 benchmarked at 1M tokens. In Figure 7, HiP + **Δ** runs more than 8 times faster. For Streaming LLM + **Δ** this factor increases to over 32, which means that **Δ** Attention may perform more than 32 attention operations for a single quadratic Flash Attention 2 operation. While our method does require more computation than the standalone sparse methods in Figure 7b, the relative increase is modest in comparison to the latency of quadratic attention. MInference has been excluded from these latency results due to the current public implementation not fully utilizing hardware parallelization in this experiment. For further details, please see Section E.

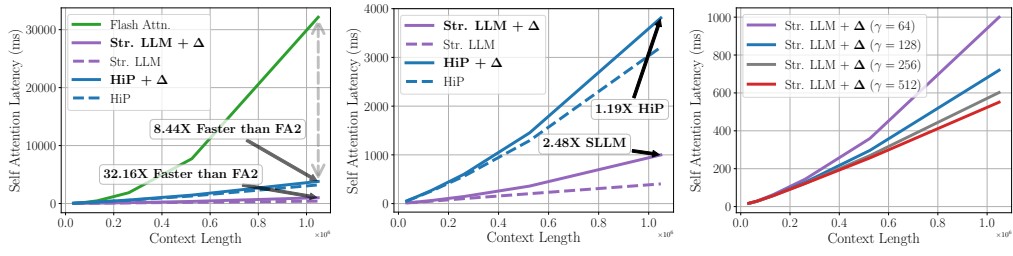

(a) Latency vs. Flash Attention     (b) Latency vs. Sparse Methods     (c) Latency for increasing $\gamma$

Figure 7: (**a**) shows latency comparisons against flash attention at 1M tokens. Our method maintains most of the large latency reductions of sparse methods. (**b**) compares latency against plain sparse methods. Our method introduces a slight overhead due to requiring computation equivalent to 1.5% of the whole attention matrix. (**c**) evaluates the effect of different $\gamma$ parameters on latency. We find that increasing the stride between queries leads to an expected decrease in latency.

**How does the $\Delta$ affect attention outputs and scores?** To study the effect of the $\Delta$ correction on the attention outputs and scores, we evaluate both attention output cosine similarity and the Spearman rank correlation coefficient [Spearman, 1904] of the attention rows for the last 128 queries of the prefill. For this, we used a sample from the MultiKey-3 RULER (131K) benchmark with the Llama3.1 8B instruction tuned model. A subset of layers is depicted in Figure 9, where each point in the plot and histogram is a random sample from one of the $32 \times 128$ (attention heads and queries). Additional plots for all layers in the network can be seen in Figures 13 to 15 in the appendix. At the key lower layers where the induction heads are known to be most prevalent, we find that the $\Delta$ correction results in a large corrective shift in both the rank correlation and cosine similarity, making both metrics much closer to the ground truth distributions of quadratic attention. Notably, only using 'Recompute', which densely recomputes some rows of the attention matrix, is not enough to shift the distribution, as it is indistinguishable from the plain Streaming LLM model in Figure 9.

In Section 1, we stated that $\Delta$ Attention shifts the distribution of attention outputs towards the distribution which would be seen under fully quadratic attention. Figure 9 provides three more examples of lower layers which show the same shift as shown in Figure 3. It is notable, however, that this strong shift towards the distribution of quadratic attention is not present in all layers of the network. Figures 13 to 15 together show all layers. $\Delta$ Attention appears to maintain a strong similarity to quadratic attention at the lower layers, which gradually dissipates until layer 10, when the three methods become indistinguishable. However, there is a sudden rise in attention output cosine similarity again towards the last layers of the network

While both the output cosine similarity and the rank correlation are important, the high rank correlation coefficient provides a crucial insight as to how the $\Delta$ correction aids in improving performance. For sparse methods, the last 128 queries from a 131K context have undergone a distributional shift induced by the sparse method, which means that they no longer correctly align with the appropriate key tokens during dot-product attention. A high rank correlation, however, implies that the ranking (importance order) of dot products across an entire row of the attention matrix remains largely intact and therefore, should result in outputs with higher similarity to quadratic attention outputs. This suggests that dense decoding can now effectively access information buried deep in the prompt, which is something our experiments show sparse attention methods struggle to do.

**Does Equation (5) or Equation (6) Perform Better?** In the previous paragraph we gave qualitative examples of the difference between Equation (5) and Equation (6) on the attention output cosine similarity. Now we ask, how does this observed difference affect the performance of the model? Table 4 shows the effect of 'recompute' from Equation (5), which

Table 4: RULER ablation for Equation (5) 'recompute' and Equation (6) $\Delta$.

| Model | 131K | 65K | 32K | … | Avg. |
|---|---|---|---|---|---|
| Str. LLM | | 27.45 | 18.59 | 30.25 | … | 44.25 |
| Str. LLM + Recompute | 52.67 | 72.71 | 78.39 | … | 79.99 |
| **Str. LLM + $\Delta$** | 64.40 | 75.22 | 81.27 | … | **83.06** |

recomputes a selected number of queries with dense attention and does not apply the difference to subsequent tokens in the $\gamma$ neighborhood. Only recomputing tokens results in a 37%pt increase over all context lengths and is only 3%pt short of matching $\Delta$. However, at the longest context length, $\Delta$ still delivers a more than 11%pt increase in accuracy.

Figure 8 shows 'recompute' compared to $\Delta$ Attention for individual subsets of the RULER-131K context length. We find that the only case where 'recompute' outperforms our method is on the

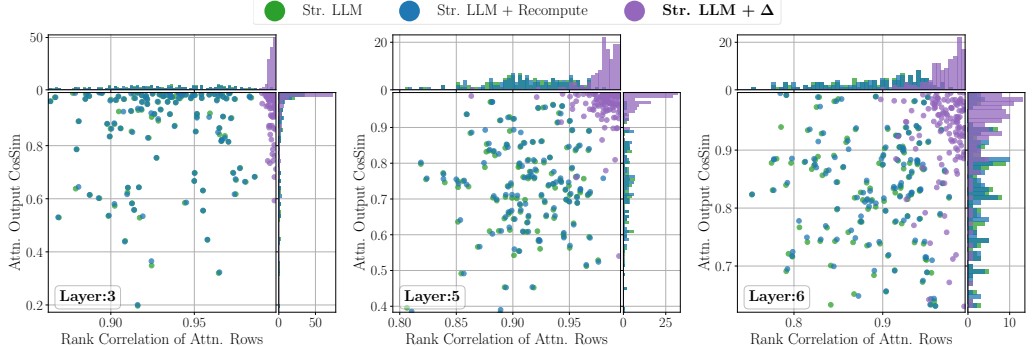

Figure 9: For RULER with a context length of 131K, we look at the final 128 tokens in the attention output and the final 128 queries in the attention matrix. We compare the cosine similarity of the outputs and the rank correlation of the attention rows to quadratic attention. We find that for both measures, $\mathbf{\Delta}$ Attention is more similar to quadratic attention.

variable tracking subset (VT). We are unsure of the cause of this anomaly, although it is important to note that 'recompute' even outperformed flash attention by approximately 15%pt, which implies that there is some structure within this task that happened to benefit from 'recompute'. In general, flash attention should represent an upper bound to sparse attention, which is what we observe in general. Note that the CWE subset of RULER is removed from this plot, as all methods (including flash attention) score 0% on the 131K context length.

## 5   Discussion & Limitations

Our method presented thus far has been a simple extension to existing sparse attention methods, which can be applied with a minimal addition of overhead and a very simple modification to the attention layer. The common way of computing sparse attention in prior work is to compute an attention output that is dense in the queries and sparse in the keys, so that there is at least one output for every input query token. One way to view our $\mathbf{\Delta}$ Attention extension is that we are mixing a key-sparse (and query-dense) attention output with a query-sparse (and key-dense) attention output in order to arrive at a representation which is closer to the quadratic attention output that is dense in both the queries and keys.

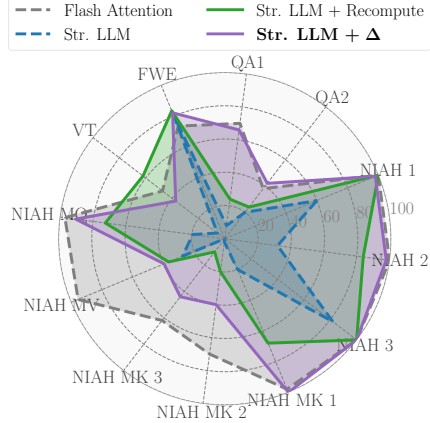

Figure 8: Comparing the effects of Equation (5) 'recompute' and Equation (6) $\mathbf{\Delta}$ on RULER 131K subsets.

The idea of viewing attention sparsity from both dimensions holds the potential for future works to explore novel ways of combining various combinations of sparse methods in order to approximate the full attention operation. With Lemma 1, we were able to show that the difference of attention outputs approximates the missing attention output, however, we only have empirical evidence of the secondary approximation that $(\mathbf{A}^{\mathbf{\Delta}}\mathbf{V})_i \approx (\mathbf{A}^{\mathbf{\Delta}}\mathbf{V})_{i+\nu}$ for $\nu \in \{1, \ldots, \gamma\}$. While this is empirically validated in our experiments and by the high cosine similarity in Figure 6b, future works may study this approximation further, which could lead to creating a smarter selection criteria for the query sparse attention, as our method uses only a fixed hyperparameter to set the size of the gap between query tokens.

## 6   Conclusion

In this work, we first diagnose a harmful distributional shift induced by sparse attention prefill methods. We then propose a remedy with our lightweight, sparse-kernel agnostic $\mathbf{\Delta}$ Attention procedure. $\mathbf{\Delta}$ Attention corrects sparse outputs to align better with full quadratic attention outputs, requiring only a small post-processing step that can be integrated seamlessly into existing inference pipelines. Across all benchmarks, and especially at the longest context lengths, our method delivers significant accuracy gains while maintaining high sparsity and low latency.

# 7 Acknowledgments

This work was supported by:

- Institute for Information & communications Technology Planning & Evaluation(IITP) grant funded by the Korea government(MSIT) (RS-2019-II190075, Artificial Intelligence Graduate School Program(KAIST))
- National Research Foundation of Korea (NRF) grant funded by the Korea government (MSIT) (No. RS-2023-00256259)
- Artificial intelligence industrial convergence cluster development project funded by the Ministry of Science and ICT(MSIT, Korea) & Gwangju Metropolitan City
- The Institute of Information & Communications Technology Planning & Evaluation (IITP) with a grant funded by the Ministry of Science and ICT (MSIT) of the Republic of Korea in connection with the Global AI Frontier Lab International Collaborative Research. (No. RS-2024-00469482 & RS-2024-00509279)
- DeepAuto R&D Program (No. DA-RS-2025-01)
- A gift grant from Google

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

## A    Appendix Contents

## B    Broader Impact

We are not aware of any negative potential impacts of our work beyond impacts that are general to all machine learning models. However, lowering the computational cost for inference has the potential to lower costs such as electricity consumption, hardware requirements, and latency for end users. If this can effectively be done with minimal degradation in the performance of the underlying model, it will likely be beneficial to both producers and consumers of AI models.

## C    Implementation Details

In addition to the index selection in Equation (4), in practice, we also select a block of queries for dense recomputation at the end of the prefill sequence, which makes the part of the prefill which requires a delta correction evenly divisible by $\gamma$. We do this for both ease of implementation and also to provide the decoding tokens with the most accurate block of recent context. The block of queries at the end of the sequence allows us to simply reshape a tensor and project the $\Delta$ correction onto every element in the block, as the tensor that needs a delta correction will have a regular size that is divisible by $\gamma$.

## D    Compute Resources

For LLM inference on benchmark datasets, we use Google Cloud Platform's 8x NVIDIA H100 node. For latency measurements, we use a standalone machine with an NVIDIA RTX 4090 in order to have a controlled environment. Here, we show the detailed specification of the latency benchmarking machine:

| | |
|---|---|
| CPU | AMD Ryzen 7950X, 16 Core, 32 Thread |
| RAM | 128GB, DDR5 5600 Mhz |
| GPU | Nvidia RTX 4090, VRAM 24GB |
| PCIe | Gen 4.0 x8 |
| OS | Ubuntu 22.04.4 LTS |
| GPU Driver | 535.171.04 |

## E    Latency of MInference with Delta Attention

We did not report the latency of MInference in the main paper, because MInference shows unusually slower latency than other tested methods, including Flash Attention. We think this is due to (1)

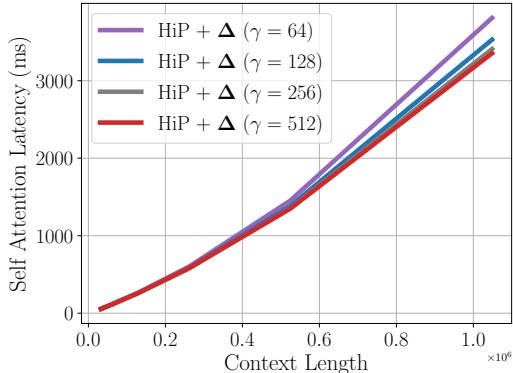

Figure 10: Latency measurements for different settings of $\gamma$ which controls the gap size between queries and also the overall sparsity of the calculation. This figure accompanies the latency ablation for Streaming LLM in the main text, Figure 7c.

insufficient optimization of the publicly available kernel [1] and (2) MInference uses a for-loop across the head dimension that prevents the head dimension from being parallelized within the GPU. This limitation of the publicly available implementation will cause the latency to suffer if the attention calculation for each head does not fully utilize the hardware. This for-loop structure was likely implemented in this way because MInference uses different sparse attention strategies for each head. Therefore, as MInference is algorithmically faster than flash attention, we do not report the latency in Figure 7, as this would be misleading to readers who are not familiar with the low-level details of the implementations.

We capture the kernel latencies and hardware utilization for MInference. In our analysis with Nsight Systems, their vertical slash pattern kernel '`_triton_mixed_sparse_attn_fwd_kernel`', shows around 32 milliseconds latency for a single head, while flash attention shows only 462 milliseconds for 32 heads. The MInference kernel shows noticeably low utilization of streaming multiprocessor warps, which is around 9%.

However, for completeness, we put the latency measurements of MInference in Table 5. In our measurement using their official codebase without meaningful modification, with pre-compiled model configuration for head-wise sparse method settings, Minference is about 1.377 times slower than Flash Attention. We believe this is only due to the lack of a fully parallelized kernel and not the design of the method.

Table 5: Prefill latency measurements (ms) that include MInference on RTX 4090 up to 256K context length.

|              | 32K    | 64K    | 128K    | 256K    |
|--------------|--------|--------|---------|---------|
| FA           | 34.27  | 119.77 | 462.39  | 1858.60 |
| HiP          | 53.61  | 118.53 | 255.05  | 562.24  |
| HiP + $\Delta$ | 55.44  | 123.49 | 268.02  | 602.74  |
| Minference   | 135.28 | 395.92 | 1083.66 | 2559.47 |

## F   Approx Window Size Calculation

When comparing out method to Streaming LLM, we would like to know how much computation overhead is increased in order to estimate the approximate window size of our method due to the fact that $\Delta$ Attention computes extra tokens. We can calculate this as follows with $C$ as the context size, and $w$ as the window size in a single row of the attention matrix, our method will

---

[1]https://github.com/microsoft/MInference

compute every $\gamma^{\text{th}}$ row of the attention matrix which would be equivalent to $\frac{C}{2\gamma}$ when amortized into each row calculation. This brings the total calculation per row to $w + \frac{C}{2\gamma}$. In the case of 131K context, a window size of 2048, and $\gamma = 64$ (our standard setting) this would be evaluated as $2048 + \frac{2^{17}}{2(2^6)} = 2048 + 2^{10} = 2048 + 1024 = 3072$.

## G   Restatement and proof of Lemma 1

We want to show that the difference of $\mathbf{A}^{\triangle}\mathbf{V} \approx \mathbf{A}\mathbf{V} - \mathbf{A}^{*}\mathbf{V}$ is approximately equal to the missing delta-shaped attention output, which is pictured in Figure 5. w.l.o.g., we will consider a single arbitrary row of the attention matrix $\mathbf{a}$ and a single column vector from the values $\mathbf{v}$. The following is true regardless of the selected entries in $\mathbf{a}$, however, in order to create a tighter error bound, we assume the existence of a sparse attention method which chooses the largest attention values in $\mathbf{a}$ when calculating the sparse dot product $\mathbf{v}^{\top}\mathbf{a}^{*}$. Specifically,

**Lemma** (Lemma 1). *Let $\bar{\mathbf{a}} = (\bar{a}_1, \ldots, \bar{a}_d) \in \mathbb{R}^d$ be the pre-softmax vector which is sorted and satisfies,*

$$\bar{a}_1 \leq \bar{a}_2 \leq \cdots \leq \bar{a}_N,$$

*then any exact top-$k$ sparse attention method which selects the top-$k$ attention scores should select the last $k$ elements of $\mathbf{a}$. Fix an integer $1 \leq k \leq N$. Define the head-sum $H$, tail-sum $T$, and normalization constant $Z$ to be the following:*

$$H = \sum_{i=1}^{N-k} e^{\bar{a}_i}, \tag{8}$$

$$T = \sum_{i=N-k+1}^{N} e^{\bar{a}_i}, \tag{9}$$

$$Z = H + T. \tag{10}$$

*Set*

$$\mathbf{a}_i = \frac{e^{\bar{a}_i}}{Z}, \qquad \mathbf{a}^{*}{}_i = \begin{cases} 0, & i \leq N - k, \\ \dfrac{e^{\bar{a}_i}}{T}, & i > N - k. \end{cases}$$

*For any $\mathbf{v} = (v_1, \ldots, v_d) \in \mathbb{R}^d$ which is sorted according to the rank of elements in $\mathbf{a}$, define the tail-max as,*

$$M_{\text{tail}} = \max_{i > N-k} |v_i|.$$

*write*

$$\boldsymbol{\Delta} = \mathbf{a}^{\top}\mathbf{v} - \mathbf{a}^{*\top}\mathbf{v},$$

*we have the exact decomposition*

$$\boldsymbol{\Delta} = \sum_{i=1}^{N-k} \mathbf{a}_i \mathbf{v}_i + R,$$

*where the "remainder" term*

$$R = \sum_{i=N-k+1}^{N} \left[\mathbf{a}_i - \mathbf{a}^{*}{}_i\right] \mathbf{v}_i$$

*is upper bounded by*

$$|R| \leq \frac{H}{H+T} M_{\text{tail}}.$$

*Therefore,*

$$\left| \boldsymbol{\Delta} - \sum_{i=1}^{N-k} \mathbf{a}_i \mathbf{v}_i \right| = |R| \tag{11}$$

$$\leq \frac{H}{H+T} M_{\text{tail}} \tag{12}$$

*Proof.* Split

$$\boldsymbol{\Delta} = \sum_{i=1}^{N-k} \mathbf{a}_i \mathbf{v}_i + \sum_{i=N-k+1}^{N} \left[\mathbf{a}_i - \mathbf{a}^*{}_i\right] \mathbf{v}_i \tag{13}$$

$$= \sum_{i=1}^{N-k} \mathbf{a}_i \mathbf{v}_i + R. \tag{14}$$

For $i > N - k$,

$$\mathbf{a}_i = \frac{e^{\bar{a}_i}}{H+T} = \frac{e^{\bar{a}_i}}{T} \frac{T}{H+T} = \mathbf{a}^*{}_i \frac{T}{H+T},$$

so

$$\mathbf{a}_i - \mathbf{a}^*{}_i = \mathbf{a}^*{}_i \frac{T}{H+T} - \mathbf{a}_i^* \tag{15}$$

$$= \mathbf{a}^*{}_i \left(\frac{T}{H+T} - 1\right) \tag{16}$$

$$= -\mathbf{a}^*{}_i \frac{H}{H+T}. \tag{17}$$

Thus

$$R = -\frac{H}{H+T} \sum_{i=N-k+1}^{N} \mathbf{a}^*{}_i \mathbf{v}_i,$$

and since $\sum_{i=N-k+1}^{N} \mathbf{a}^*{}_i = 1$ and $|\mathbf{v}_i| \leq M_{\text{tail}}$ on the tail,

$$|R| = \frac{H}{H+T} \left| \sum_{i=N-k+1}^{N} \mathbf{a}^*{}_i \mathbf{v}_i \right| \tag{18}$$

$$\leq \frac{H}{H+T} \sum_{i=N-k+1}^{N} \mathbf{a}^*{}_i |\mathbf{v}_i| \tag{19}$$

$$\leq \frac{H}{H+T} M_{\text{tail}}. \tag{20}$$

completing the proof. $\qquad\square$

If we assume that $T \gg H$ as is the expected outcome with sparse attention, then the bound becomes tighter, as the denominator $H + T \gg H$. This implies that better sparse top-$k$ approximations will result in a lower error bound. We empirically verified this difference in Figure 11, which analyzes both the error bound and the empirical error on a real input from the RULER-131K subset. Figure 11a measures the bound and empirical error of an oracle top-$k$ attention while Figure 11b measures the same bound and empirical error for Streaming LLM, which chooses a sliding window and attention sink. We find that the bound is generally tighter for the oracle top-$k$ attention, but in both cases, the overall empirical approximation error remains low.

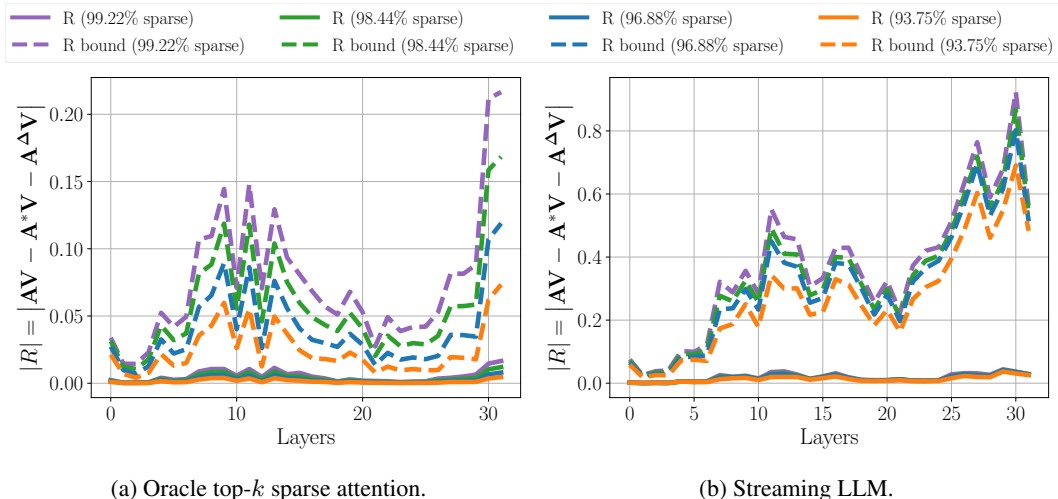

(a) Oracle top-$k$ sparse attention.

(b) Streaming LLM.

Figure 11: Empirically analyzing the approximation and bound from Lemma 1. A more precise sparse top-$k$ attention method, such as an oracle (a) maintains a tighter bound on the approximation error. Streaming LLM (b) results in a looser bound, however the empirical approximation error (solid lines) remains low in both methods.

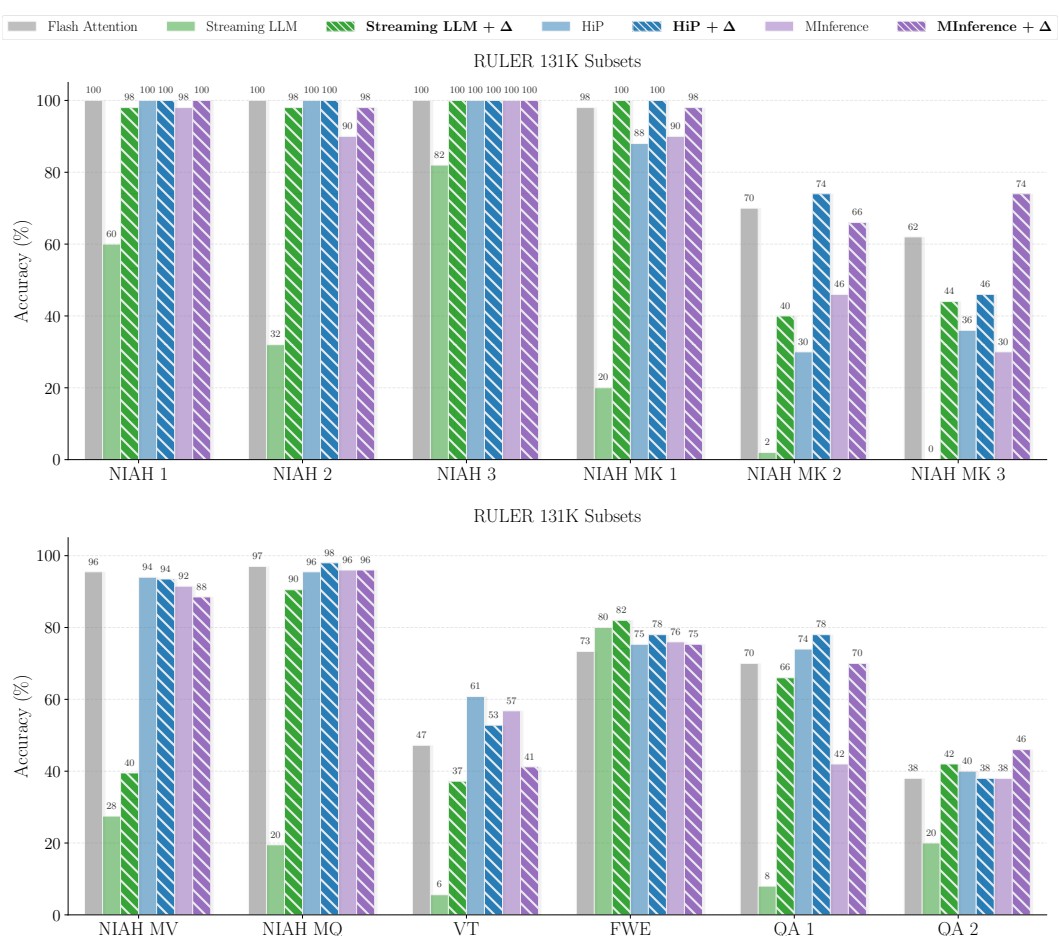

Figure 12: All RULER 131K subsets. This is a companion to Figure 1. The CWE subset is excluded, as all models, including quadratic attention, scored 0%.

## H  Extended Related Work

In addition to the related work cited in Section 2, there are a number of additional works which deal with related topics that we wish to highlight.

LESS [Dong et al., 2024] requires training a low rank cache compressor. LESS mentions differences in attention distributions between dense and sparse attention, however, the authors make no mention of the critical insight of our work, namely that a dense decode fails to properly align with the tokens resulting from a sparse prefill due to the distributional shift of the keys that is induced by the sparse prefill.

Cacheblend [Yao et al., 2024] proposes using one dense attention layer to identify important tokens, and then selectively recomputing these tokens in later layers to add missing parts of the sparse attention to cached KV pairs. Cacheblend proposed this as a way to augment and consolidate independently processed chunks of a RAG pipeline. In practice, however, this would effectively be similar to a "smart" sparse prefill method like HiP or MInference which fills in some of the missing tokens in the attention matrix which are outside of the local window context. As out experiments show, this is not always sufficient to fix the distributional shift between sparse and dense prefills.

APE [Yang et al., 2025] proposes temperature scaling and rescaling the attention post-hoc in order to correct any error introduced. However, APE misses the crucial insight of our work, namely that sparse and dense attention result in completely different token distributions which means that there is a problem of query-key matching during decoding. APE only considers query and key geometry as a function of a) position and b) input. They deduce that because they key states of the first few keys (sink tokens) are relatively stable, then the geometry of all other keys are also stable.

Rectified Sparse Attention [Sun et al., 2025] (a concurrent work) considers dense prefills and a sparse decoding procedure. Their sparse decoding procedure is similar to what we call "recompute" in Table 4 and Figures 8, 9 and 13 to 15 where we showed that this "recomptue" method is insufficient to mitigate the distributional shift in the outputs.

Comparisons to these extended related works, and to Star Attention [Acharya et al., 2024] can be seen in Table 6.

Table 6: RULER comaprison to related works on sparse attention and sparse RAG works.

| Method | 131K | 65K | 32K | 16K | 8K | 4K | Avg. |
|---|---|---|---|---|---|---|---|
| Str.LLM | 27.45 | 18.59 | 30.25 | 38.13 | 60.53 | 90.52 | 44.25 |
| Cachblend | 0.00 | 0.21 | 0.31 | 1.49 | 24.42 | 96.27 | 20.45 |
| APE | 26.76 | 43.03 | 53.13 | 67.50 | 77.25 | 93.76 | 60.24 |
| Str.LLM + Delta | **64.40** | **75.22** | **81.27** | **88.66** | **92.25** | **96.54** | **83.06** |
| Star Attention Mask | 12.00 | 14.86 | 20.43 | 31.66 | 51.60 | 78.62 | 34.86 |
| Star Attention Mask + Delta | **58.84** | **70.12** | **74.77** | **82.69** | **89.12** | **93.28** | **78.13** |

## I  RepoQA

We evaluate $\Delta$ Attention on code understanding by using the RepoQA [Liu et al., 2024] dataset that asks the model to retrieve a function from a long block of input text. In this dataset, the long input text contains the code from many functions and the query contains a plain language description of what the function does. The model is then supposed to return back the correct function as output. We compare Streaming LLM with and without our delta correction in Table 7

## J  Interpolation

The method presented in Section 3 proposes to use a single delta correction at index $i$ to influence the next $i + \gamma - 1$ attention outputs. This causes a discrete jump in the delta correction at every $\gamma^{\text{th}}$ output. It may be the case that a better strategy would be to smooth out the transition or impute the delta corrections within the window by some imputation function. In Table 8, we look at three different possible imputation functions and evaluate the overall effect on RULER.

Table 7: RepoQA results for Streaming LLM and Streaming LLM + Delta. Plain FA3 is included for reference.

| Threshold | 0.0 | 0.1 | 0.2 | 0.3 | 0.4 | 0.5 | 0.6 | 0.7 | 0.8 | 0.9 | 1.0 | Avg |
|---|---|---|---|---|---|---|---|---|---|---|---|---|
| Vanilla (FA3) | 94.8 | 92.2 | 90.6 | 89.4 | 88.8 | 88.4 | 86.8 | 85.4 | 84.4 | 83.2 | 76 | 87.27 |
| Str.LLM | 73.6 | 64.0 | 60.6 | 58.8 | 57.6 | 56.8 | 55.0 | 53.0 | 50.4 | 44.2 | 35.6 | 55.42 |
| Str.LLM + Delta | **85.8** | **78.0** | **73.8** | **72.0** | **70.6** | **67.0** | **64.8** | **61.2** | **57.2** | **50.4** | **42.6** | **65.76** |

---

Algorithm 2: $\alpha, \beta, \gamma$ Filter

---

**Require:** $\alpha, \beta, \gamma$ and $\Delta$ vectors
   $o \leftarrow$ zero vector like $\Delta$
   $p \leftarrow \Delta_0$
   $v \leftarrow$ zero vector like $p$
   $a \leftarrow$ zero vector like $p$
   $o \leftarrow \Delta_0$
   **for** $i$ in $[1, ..., \text{len}(\Delta)]$ **do**
      $y \leftarrow \Delta_i$
      // update approx position and velocity
      $\hat{p} \leftarrow p + v + 0.5a$
      $\hat{v} \leftarrow v + a$
      // calculate difference between real and predicted position
      $r \leftarrow y - \hat{p}$
      // update position, velocity, and acceleration.
      $p \leftarrow \hat{p} + \alpha r$
      $v \leftarrow \hat{v} + \beta r$
      $a \leftarrow a + \gamma r$
      $o_i \leftarrow p$
   **end for**
   return $o$

---

**Linear Interpolation.** For linear interpolation, we first compute all delta corrections, and then produce mixing coefficients $\beta \in [0, 1]$ which linearly increase from $[0, ..., 1]$. Interpolation is then performed between consecutive delta correction terms by the function $\hat{\Delta}_k = (1 - \beta_k)\Delta_i + \beta_k \Delta_{i+1}$. Each $\Delta$ term will therefore expand into $|k| = \gamma$ terms, such that the number of delta corrections now matches the sparse attention output size. These expanded, and smoothed delta corrections will be treated as the new correction term, providing a smoother transition between terms.

**EMA.** Instead of linear interpolation, which technically violates the causality of the attention mechanism by incorporating information from the future into the past, we may instead expand the delta correction term by repeating each vector $\gamma$ times, and then perform an exponential moving average (EMA) over the full set of vectors using a coefficient $\beta \in [0, 1]$ and computing the EMA as $\Delta_i = (1 - \beta)\Delta_{i-1} + \beta\Delta_i$. The EMA acts as a smoothing mechanism which smooths the transition between delta terms.

$\alpha, \beta, \gamma$ **Filter.** A third option is to use a Kalman style filter. We chose to use an $\alpha, \beta, \gamma$ filter where $\alpha$ is a position coefficient, $\beta$ is a velocity coefficient, and $\gamma$ is an acceleration coefficient. At each step, position, velocity, and acceleration are updated based on a mixture of the real position and the accumulated statistics for position, velocity, and acceleration. We consider every operation to be an elementwise scalar operation. The algorithm for the $\alpha, \beta, \gamma$ filter can be seen in Algorithm 2

Although there are slight improvements using these imputation methods in Table 8, no method shows conclusive improvements over our original method. However, we think delta smoothing or imputation shows a promising direction for future research.

Table 8: Interpolation Experiments.

| Method | 131K | 65K | 32K | 16K | 8K | 4K | Avg. |
|---|---|---|---|---|---|---|---|
| Str.LLM + Delta + Linear Interpolation | **65.15** | **75.65** | 81.26 | 88.26 | 92.34 | **96.66** | **83.22** |
| Str. LLM + Delta + EMA ($\beta = 0.5$) | 63.21 | 75.22 | **81.27** | **88.66** | 92.25 | 96.54 | 82.85 |
| Str. LLM + Delta + EMA ($\beta = 0.75$) | 63.40 | 74.60 | 80.76 | 88.52 | 92.29 | 96.62 | 82.69 |
| Str. LLM + Delta + EMA ($\beta = 0.95$) | 63.16 | 75.87 | 81.03 | 88.27 | 92.26 | 96.59 | 82.86 |
| Str. LLM + Delta + ($\alpha = 0.05, \beta = 1.25 \times 10^{-4}, \gamma = 2.08 \times 10^{-5}$) Filter | 58.35 | 73.48 | 80.54 | 88.15 | 92.03 | 96.48 | 81.50 |
| Str. LLM + Delta + ($\alpha = 0.1, \beta = 5 \times 10^{-3}, \gamma = 1.66 \times 10^{-4}$) Filter | 57.99 | 72.70 | 79.64 | 88.58 | **92.42** | 96.57 | 81.31 |
| Str. LLM + Delta + ($\alpha = 0.2, \beta = 5 \times 10^{-2}, \gamma = 3.5 \times 10^{-3}$) Filter | 61.47 | 74.31 | 80.29 | 88.47 | 92.32 | 96.58 | 82.24 |
| Str.LLM + Delta | 64.40 | 75.22 | **81.27** | **88.66** | 92.25 | 96.54 | 83.06 |

# K   Statistical Significance Tests

We assess the statistical significance of the results presented in Table 1. For this, we use a one-sided paired permutation test to test the significance of the difference between the versions of Streaming LLM, HiP and MInference with and without our delta correct applied. The results are shown in Table 9. We split RULER tasks according to QA vs. non-QA retrieval tasks. The statistical significance shows a high correlation with the displayed colors in Table 1 and verifies that our results are statistically significant.

Table 9: Interpolation Experiments. Each entry is a p-value assessing whether or not our delta correction results in a significant improvement (significance level is $p < 0.05$).

| Method | 131K | 65K | 32K | 16K | 8K | 4K |
|---|---|---|---|---|---|---|
| Str.LLM (all non qa tasks) | **0.0001** | **0.0001** | **0.0001** | **0.0001** | **0.0001** | **0.0001** |
| Str.LLM (all qa tasks) | **0.0001** | **0.0001** | **0.0001** | **0.0001** | **0.0001** | 0.4958 |
| HiP (all non qa tasks) | **0.0001** | **0.0018** | 0.7112 | 0.5018 | 0.8331 | 0.5747 |
| HiP (all qa tasks) | 0.4918 | 0.7252 | 0.9245 | 0.8076 | 0.5009 | 1 |
| MInference (all non qa tasks) | **0.0001** | 0.858 | 0.6485 | 0.5116 | 0.8774 | 1 |
| MInference (all qa tasks) | **0.0004** | 0.8813 | 0.9848 | 0.8777 | 0.499 | 1 |

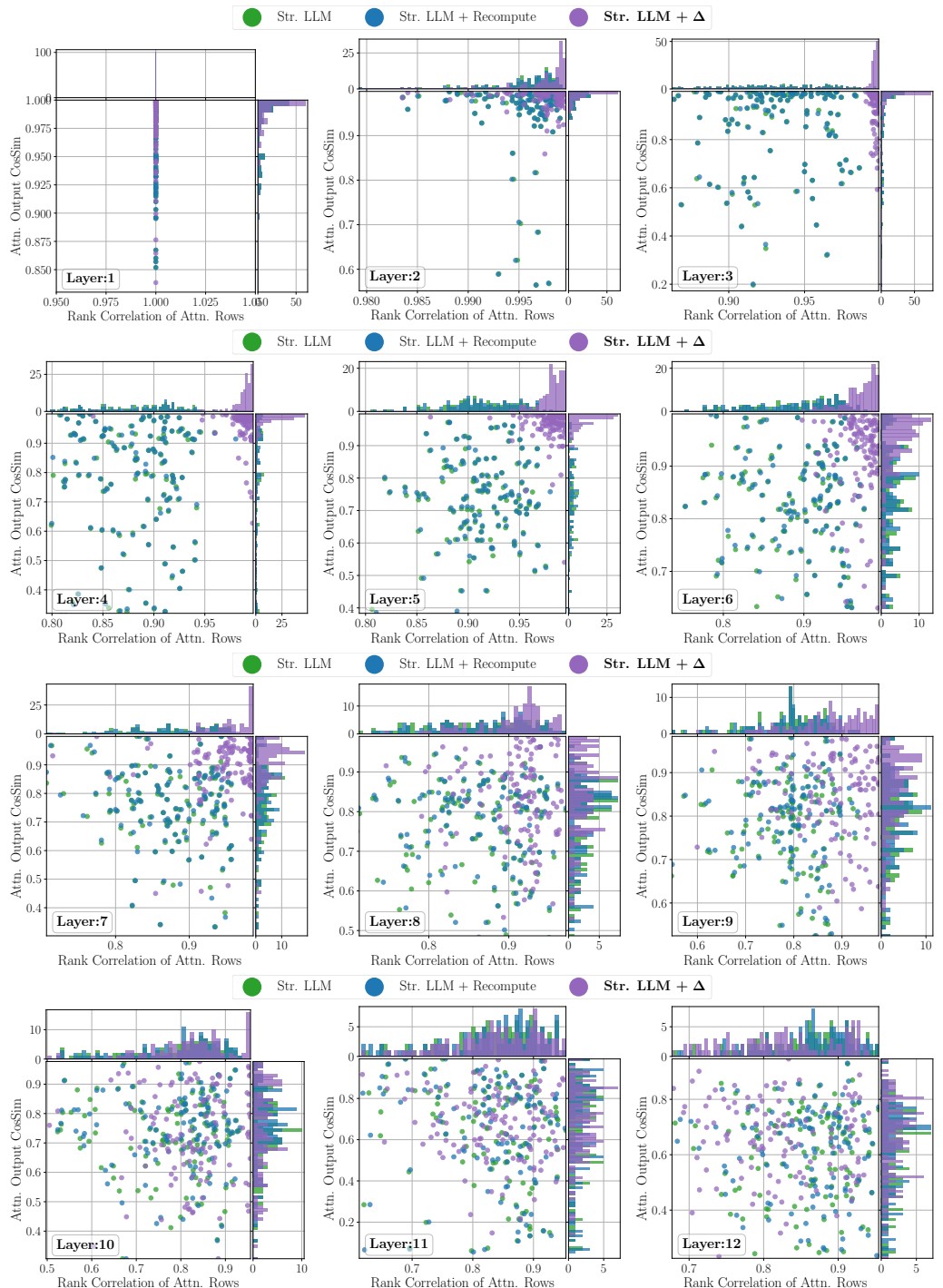

Figure 13: Attention output cosine similarity (compared to full attention) for Streaming LLM with our method. Figures 13 to 15 show the results from every layer, and are a counterpart to Figure 9 in the main text. For the lower layers where induction heads are most prevalent, our method shows higher cosine similarity and attention row rank correlation as compared to quadratic attention.

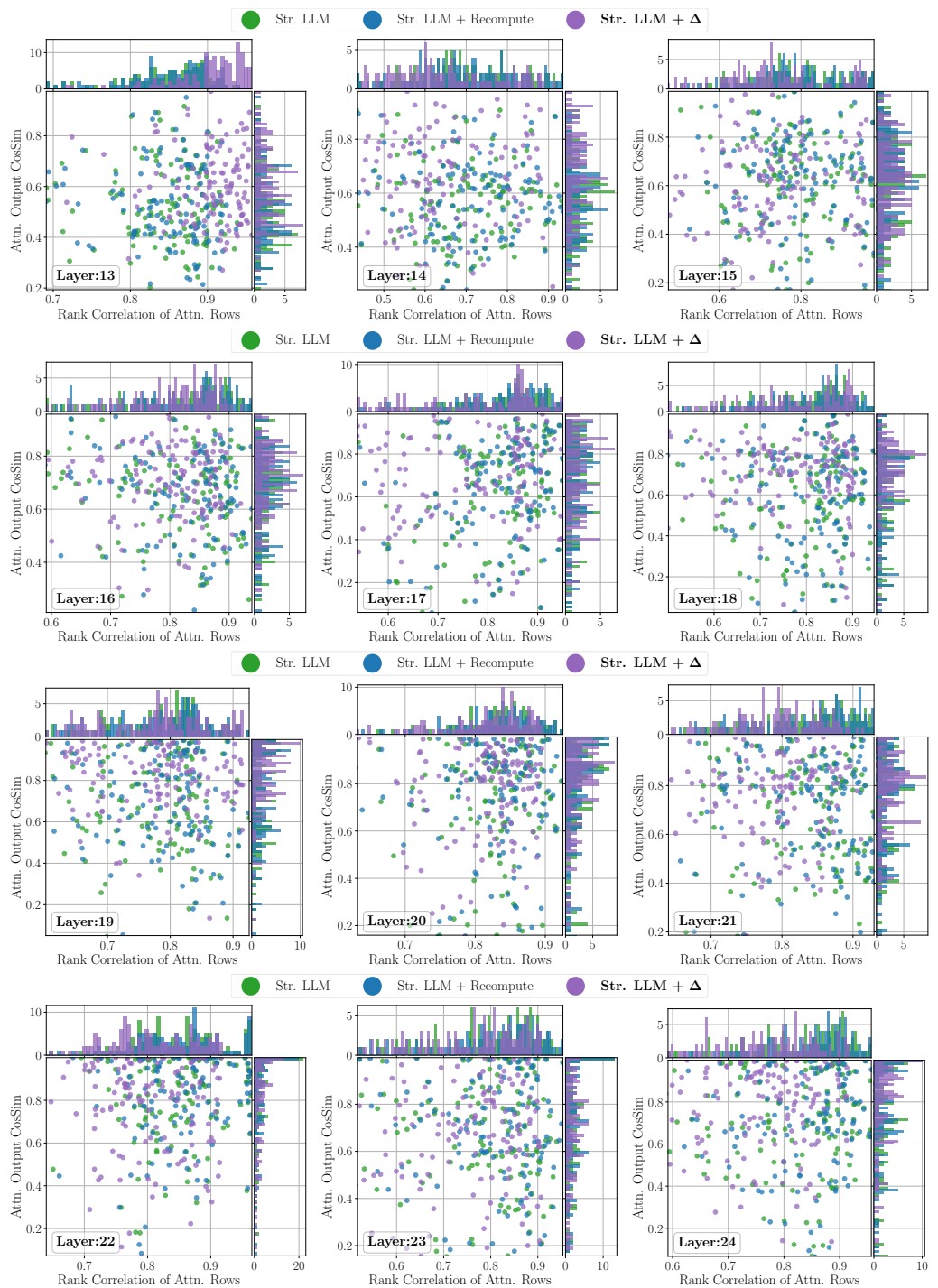

Figure 14: Attention output cosine similarity (compared to full attention) for Streaming LLM with our method. Figures 13 to 15 show the results from every layer, and are a counterpart to Figure 9 in the main text. For the lower layers where induction heads are most prevalent, our method shows higher cosine similarity and attention row rank correlation as compared to quadratic attention.

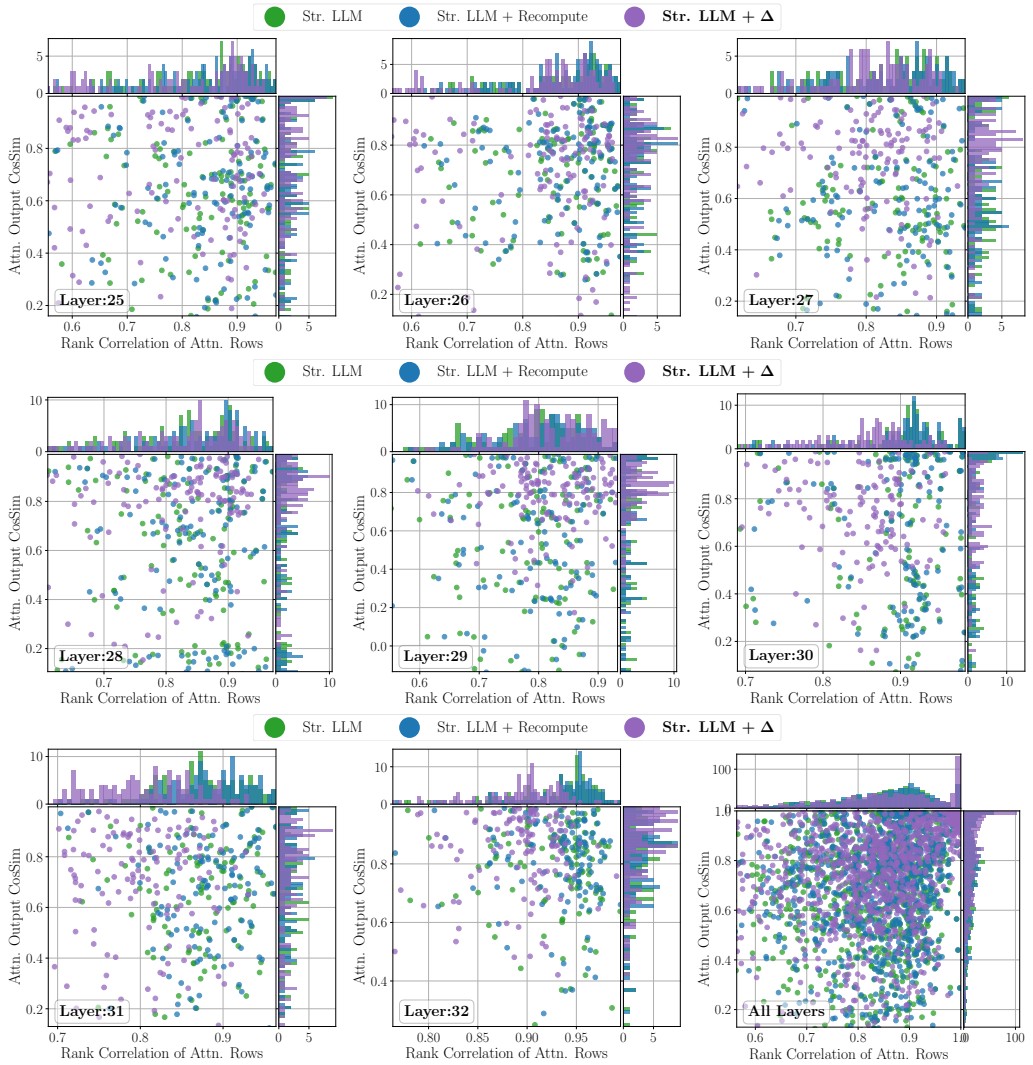

Figure 15: Attention output cosine similarity (compared to full attention) for Streaming LLM with our method. Figures 13 to 15 show the results from every layer, and are a counterpart to Figure 9 in the main text. For the lower layers where induction heads are most prevalent, our method shows higher cosine similarity and attention row rank correlation as compared to quadratic attention.

