# OpenReview forum: "Delta Attention: Fast and Accurate Sparse Attention Inference by Delta Correction"
_NeurIPS.cc/2025/Conference — NeurIPS 2025 poster_

### Official Review · Reviewer_wJZr · 2025-06-30

**Clarity:** 3
**Significance:** 2
**Originality:** 2
**Rating:** 4
**Confidence:** 4

**Summary:**

- This work proposes a new sparse attention method targeting the computational complexity of self-attetion during the model inference.
- This method exploits the distribution shift between the full attention and sparse attention, and proposes a delta method to bridge this distribution gap.
- This method is extensively evaluated on several long context tasks, achieving superior performances than existing methods while keeping sparsity.

**Questions:**

- i wonder does it requires task-specific tuning for ultra-long contexts.
- the proposed method seems rely on an assumption that nearby tokens have similar representations, this may not be the case in complex or reasoning tasks, while the proposed method is mostly evaluated on long context benchmarks.

**Ethical Concerns:**

["NO or VERY MINOR ethics concerns only"]

**Limitations:**

Please refer to the weakness and questions

**Quality:**

2

**Strengths And Weaknesses:**

# strength
- The paper is well written and the motivation is clear. the analysis about the distribution shift of attention scores shown in fig 3 is insightful.
- The performance gains seem significant, while the increased latency is minimum. the ablations about $\Delta$ and recomputation are comprehensive.
- As reported, this method could be seamlessly integrated with existing sparse methods.
# weakness
- Ablation of $\gamma$ on down-stream tasks are preferred to be reported (now only ppl). i wonder does it requires task-specific tuning for ultra-long contexts.
- Correction efficacy diminishes in higher network layers (Appendix Figs. 13-15), which may limit the gains in more complex and reasoning tasks.

---

> ### Author Rebuttal · Authors · 2025-07-30
>
> Thank you for your time and effort in reviewing our work and highlighting the positive attributes of our method. We will address each of your concerns in turn below.
>
> ---
>
> > I wonder does it requires task-specific tuning for ultra-long contexts.
>
> We did no task specific tuning for ultra long contexts, except for the reasoning experiment listed below. $\gamma$ is the only tunable parameter in delta attention. We uniformly chose $\gamma = 64$ in our original submission for all experiments except for ablations on $\gamma$.
>
> ---
>
> > Correction efficacy diminishes in higher network layers (Appendix Figs. 13-15), which may limit the gains in more complex and reasoning tasks. | the proposed method seems rely on an assumption that nearby tokens have similar representations, this may not be the case in complex or reasoning tasks
>
> Thank you for the insightful comment. To evaluate the method’s effectiveness on reasoning tasks involving long contexts, we conducted additional experiments on RepoQA, a challenging benchmark for repository-level code question answering that requires understanding long-range dependencies (e.g., reasoning about function-class relationships). In this experiment, we reduced $\gamma$ to 16 to improve correction efficacy. Importantly, even at $\gamma = 16$, the compute overhead of the query sparse attention remains at 1/16 that of Flash Attention. Our method demonstrates strong empirical performance on this task, suggesting that it remains effective even in complex reasoning scenarios.
>
> Note that in this task, the threshold setting in the top row is a similarity threshold for score matching (0.8 is default).
>
> **Table: RepoQA with Qwen3 235B A22B Reasoning Activated**
>
>
> Threshold | 0.0 | 0.1 | 0.2 | 0.3 | 0.4 | 0.5 | 0.6 | 0.7 | 0.8 | 0.9 | 1.0 | Avg
> -- | -- | -- | -- | -- | -- | -- | -- | -- | -- | -- | -- | --
> Str.LLM | 73.6 | 64 | 60.6 | 58.8 | 57.6 | 56.8 | 55 | 53 | 50.4 | 44.2 | 35.6 | 55.42
> Str.LLM + Delta | **85.8** | **78** | **73.8** | **72** | **70.6** | **67** | **64.8** | **61.2** | **57.2** | **50.4** | **42.6** | **65.76**
>
>
> ---
>
> Thank you again for your time and care in reviewing our work. We have done our best to address your concerns within the allotted time period. We will incorporate any new results into the final version of our work. We remain open to further discussion throughout the end of the discussion period.

---

### Official Review · Reviewer_1VjR · 2025-07-01

**Clarity:** 4
**Significance:** 3
**Originality:** 3
**Rating:** 5
**Confidence:** 4

**Summary:**

This work proposes $\Delta$ Attention, a method for mitigating the bias caused by sparse attention. The proposed method computes the residual between full attention and sparse attention on a subset of queries, and subsequently applies them as residuals for all queries under sparse attention. Empirically the proposed method improves performance for several sparse attention methods on 3 datasets.

**Questions:**

Please see my interpolation comment. Did the authors try alternate interpolation methods?

**Ethical Concerns:**

["NO or VERY MINOR ethics concerns only"]

**Final Justification:**

Updated my score after the new experiments.

**Limitations:**

yes

**Quality:**

3

**Strengths And Weaknesses:**

Strengths:
- Interesting new method which improves quality for sparse attention models.
- Extensive empirical results.
- Clear presentation.

Weakness:
- While the experiments span multiple base models and datasets, there is no statistical significance test. Statistical significance tests are useful even for deterministic methods. I'd like to encourage the authors report standard test results such as paired permutation tests.
- The writeup could benefit from analysis of more sophisticated interpolation methods. In Eq 7 the same correction is applied for contiguous $\gamma$ rows. But what if we linearly interpolate between corrections in these $\gamma$ rows, for example?

---

> ### Author Rebuttal · Authors · 2025-07-30
>
> Thank you for your time and effort in reviewing our work and highlighting that our method improves upon sparse attention methods with clear and extensive empirical results. We will address each of your concerns in turn below.
>
> ---
>
> >  There is no statistical significance test. I'd like to encourage the authors report standard test results such as paired permutation tests.
>
> We performed a paired permutation test for the RULER experiments. As QA tasks and Non-QA tasks use different evaluation metric functions for correctness, we have split the test by QA tasks and non QA tasks. We have highlighted all results which surpass the one-sided significance threshold of 0.05. The results match our intuitions about which results are significant in the text (noted by color gradient in the submission). Our method shows significant improvements over Str.LLM at all context lengths and a significant improvement over MInference and HiP at the longest context lengths.
>
> |  | 131K | 65K | 32K | 16K | 8K | 4K |
> |---|---|---|---|---|---|---|
> | Str.LLM (all non qa tasks) | **0.0001** | **0.0001** | **0.0001** | **0.0001** | **0.0001** | **0.0001** |
> | Str.LLM (all qa tasks) | **0.0001** | **0.0001** | **0.0001** | **0.0001** | **0.0001** | 0.4958 |
> | HiP (all non qa tasks) | **0.0001** | **0.0018** | 0.7112 | 0.5018 | 0.8331 | 0.5747 |
> | HiP (all qa tasks) | 0.4918 | 0.7252 | 0.9245 | 0.8076 | 0.5009 | 1 |
> | MInference (all non qa tasks) | **0.0001** | 0.858 | 0.6485 | 0.5116 | 0.8774 | 1 |
> | MInference (all qa tasks) | **0.0004** | 0.8813 | 0.9848 | 0.8777 | 0.499 | 1 |
>
> Please note that due to a unexpected server failure, the machine which contained the HiP prediction files from our original submission was lost, therefore we had to re-run them which requires RULER to regenerate the dataset and therefore there are slight deviations in the HiP results from the original submission, but no significant differences. These are the HiP results averaged over all tasks which contributed to the HiP p-values above. We will update them in the final revision of our work.
>
> |  | 131K | 65K | 32K | 16K | 8K | 4K |
> |---|---|---|---|---|---|---|
> | HiP (rerun) | 68.74 | 82.51 | 89.92 | 94.10 | 94.56 | 96.80 |
> | HiP + Delta (rerun) | 73.71 | 84.89 | 89.39 | 93.86 | 94.43 | 96.31 |
>
> ---
>
> > The writeup could benefit from analysis of more sophisticated interpolation methods.
>
> We tried this experiment with linear interpolation between delta correction values. We find a small but statistically insignificant improvement overall. Additionally, the interpolation poses a theoretical problem in that it violates the causality of the causal attention mask. Interpolation between neighboring delta corrections would introduce influence from the $i + \gamma^{\text{th}}$ row of the attention matrix into the $i^{\text{th}}$ output which is why we decided to repeat the attention rows instead of interpolating in our original work.
>
> |  | 131K | 65K | 32K | 16K | 8K | 4K | Avg. |
> |---|---|---|---|---|---|---|---|
> | Str.LLM + Delta | 64.40 | 75.22 | 81.27 | 88.66 | 92.25 | 96.54 | 83.06 |
> | Str.LLM + Delta (smooth) | 65.15 | 75.65 | 81.26 | 88.26 | 92.34 | 96.66 | 83.22 |
> | p-value (all non-qa tasks) | 0.1708 | 0.5185 | 0.3818 | 0.4279 | 0.5782 | 0.0001 | - |
> | p-value (all qa tasks) | 0.766 | 0.2455 | 0.8698 | 1 | 0.5043 | 1 | - |
>
> ---
>
> Thank you for your time and effort in reviewing our work. We will add these new results into the final version of our work. We sincerely hope we have addressed your concerns and remain open to further discussion throughout the discussion period.

---

> > ### Comment · Reviewer_1VjR · 2025-08-05
> >
> > Thanks for the rebuttal. I appreciate the statistical analyses and the additional interpolation experiments.
> >
> > > the interpolation poses a theoretical problem in that it violates the causality of the causal attention mask.
> >
> > I think It might be possible to design a regression scheme that obeys a causal constraint by only considering left-side values.

---

> ### Author Response · Authors · 2025-08-06
>
> > I think It might be possible to design a regression scheme that obeys a causal constraint by only considering left-side values.
>
> Thank you for your suggestion. This is an interesting idea, as we were originally thinking of interpolation as a way of minimizing the gap between tokens. Left side only interpolation would essentially increase the gap between tokens since the closest token on the left side is the last $\gamma^{\text{th}}$ token. As we see it, the upside may be that although we are bringing in "older" information, it would provide a smoother transition between delta corrections rather than a discrete jump in delta correction at every $\gamma^{\text{th}}$ token.
>
> For this, we think that an exponential moving average ($\hat{x}_i = (1 - \beta) x_i + \beta \hat{x}_j$ for $\beta \in [0, 1]$ and $j = i - 1$) over the tokens (after repeating the delta correction in the $\gamma$ window) would provide the proper left-side only interpolation between the delta correction terms. As this was straightforward to try, we performed an experiment on RULER to test out left side only interpolation and achieved the following results.
>
> We used 3 EMA coefficients (0.5, 0.75, 0.95) to test any difference between faster/slower transitions.
>
> **[CORRECTED EMA TABLE CAN BE SEEN BELOW]**
>
> We found that the left side only interpolation gave a slight increase in performance for the $\beta=0.95$ coefficient, but it did not pass the threshold of statistical significance when comparing to our base model (top row). Surprisingly, $\beta \in \{0.5, 0.75\}$ seems to not have changed any predictions in the model, as the accuracies are identical.
>
> Thank you for your comment and suggesting this experiment, as we did not think about left-side interpolation originally. We will add these new results to the final version of our work. We remain available for further discussion until the end of the discussion period.

---

> > ### Comment · Reviewer_1VjR · 2025-08-07
> >
> > Thank you for the additional experiments.
> >
> > To generalize your EMA smoothing scheme, one possibility might be to see the imputation problem as a state estimation smoothing task, which gives you both classical (_e.g._, Kalman filters) and NN options for more sophisticated regression algorithms.

---

> ### Author Response · Authors · 2025-08-08
>
> Thank you for this suggestion. Upon trying to implement a simple ($\alpha,\beta,\gamma$)-filter we found a bug in our EMA implementation which rendered it almost inert. In fixing it, we have found that Pytorch level iterative operations which scan over the whole sequence (up to 131K in RULER) are too slow to be practically useful (~7000ms per layer). The only way we were able to implement something fast enough for practical use is by writing a kernel-level EMA and $\alpha,\beta,\gamma$-filter which runs in ~50ms per layer (for 131K). We report both results below.
>
> **EMA Kernel**
>
> |  | 131K | 65K | 32K | 16K | 8K | 4K | p-value (qa, 131K) | p-value (non-qa, 131K) |
> |---|---|---|---|---|---|---|---|---|
> | Str. LLM + Delta | 64.40 | 75.22 | 81.27 | 88.66 | 92.25 | 96.54 | - | - |
> | Str. LLM + Delta (EMA 0.5) | 63.21 | 75.22 | 81.27 | 88.66 | 92.25 | 96.54 | 0.8916 | 0.9362 |
> | Str. LLM + Delta (EMA 0.75) | 63.40 | 74.60 | 80.76 | 88.52 | 92.29 | 96.62 | 0.6689 | 0.9701 |
> | Str. LLM + Delta (EMA 0.95) | 63.16 | 75.87 | 81.03 | 88.27 | 92.26 | 96.59 | 0.6808 | 0.9816 |
>
> We chose an ($\alpha,\beta\gamma$)-filter as it is a simplified, special case of the more general Kalman filter. It's a third-order filter, meaning it estimates position, velocity, and acceleration, and also predicts future values.
>
> **($\alpha,\beta,\gamma$)-filter Kernel**
>
> |  | 131K | 65K | 32K | 16K | 8K | 4K | p-value (qa, 131K) | p-value (non-qa, 131K) |
> |---|---|---|---|---|---|---|---|---|
> | Str. LLM + Delta | 64.40 | 75.22 | 81.27 | 88.66 | 92.25 | 96.54 | - | - |
> | Str. LLM + Delta ($\alpha0.05,\beta=0.000125,\gamma=2.08x10^{-5}$) | 58.35 | 73.48 | 80.54 | 88.15 | 92.03 | 96.48 | 0.985 | 0.9946 |
> | Str. LLM + Delta ($\alpha=0.1,\beta=0.005,\gamma=1.66x10^{-4}$) | 57.99 | 72.70 | 79.64 | 88.58 | 92.42 | 96.57 | 1 | 0.9997 |
> | Str. LLM + Delta ($\alpha=0.2,\beta=0.05,\gamma=3.5x10^{-3}$) | 61.47 | 74.31 | 80.29 | 88.47 | 92.32 | 96.58 | 1 | 0.9883 |
>
> For all three types of interpolation (causality breaking linear interpolation, EMA and ($\alpha,\beta,\gamma$)-filters), we have seen that the performance is not improved. However, we agree with the intuition that discrete jumps in the delta correction are a likely place to look for improvement and represent a promising direction for future work. We hope that our discovery of the distribution shift and delta correction will inspire others to experiment with more sophisticated imputation methods, especially those which may learn the imputation function.
>
> Thank you for suggesting these experiments. As the discussion period is coming to a close today, we hope we have fully addressed your comments. We feel these three additional experiments have contributed to our work, and we will add these new results to the final text.

---

### Official Review · Reviewer_7eBZ · 2025-07-03

**Clarity:** 4
**Significance:** 2
**Originality:** 3
**Rating:** 4
**Confidence:** 3

**Summary:**

This work aims to bridge the gap between attention scores obtained from sparse attention methods and those from full attention during the prefilling stage. The core idea is to introduce a $\Delta$ matrix that approximates the difference between sparse and full attention scores. To construct this $\Delta$ matrix, the authors compute full attention for a small subset of queries (e.g., 1/64) and assume that neighboring queries exhibit similar delta values. Based on this assumption, the $\Delta$ values from the small subset are repeated across neighboring positions to approximate the full matrix.

The proposed $\Delta$ attention mechanism adds negligible latency compared to standard sparse attention methods, while providing clear performance improvements.

**Questions:**

- Do the $\Delta$ matrices look similar across different inputs? In other words, to what extent is the $\Delta$ matrix query-dependent versus being a property of the model itself?

**Ethical Concerns:**

["NO or VERY MINOR ethics concerns only"]

**Final Justification:**

I'd like to keep my positive rating.

**Limitations:**

Please see Weaknesses.

**Quality:**

3

**Strengths And Weaknesses:**

**Strengths**
- The idea of using a small number of full attention computations to correct or enhance sparse attention scores is novel and interesting.
- The experimental results effectively demonstrate the benefits of the approach.
- The paper is well-written and easy to follow.

**Weaknesses**
1. A key assumption in the method is that delta values for neighboring queries are similar. However, this assumption is not empirically validated. It would strengthen the paper if the authors provided an analysis of the actual $\Delta$ matrix, such as measuring correlations between neighboring delta values.
2. Such an analysis could also inspire better methods for constructing the approximated $\Delta$ matrix. For instance, instead of simply duplicating delta values, interpolation techniques might yield more accurate approximations. It's unclear how performance scales with the parameter $\gamma$ (i.e., the fraction of queries used for full attention). At what $\gamma$ value does the method match the performance of full attention?

---

> ### Author Rebuttal · Authors · 2025-07-30
>
> Thank you for your time in reviewing our work and noting that our method is novel and interesting, effective, and easy to follow. We will address each concern you raise in turn below.
>
> ---
>
> > 1. A key assumption is that delta values for neighboring queries are similar … not empirically validated. …
>
> If we understand the concern correctly, we have provided this experiment in Figure 6b. In this case we study the average cosine similarity between $ (A^\Delta V)_i $ and $(A^\Delta V)_j$ for $j \in [i+1, \dots, i+\gamma]$ (notation changed from paper due to math rendering issue in OpenReview). This means that we computed the full attention output minus the sparse attention output and evaluated the average cosine similarity for different gamma windows. In Figure 6b we find a generally high cosine similarity for all delta windows, although it does show a tendency to decrease for larger windows. This high cosine similarity implies that the delta may be re-used over multiple output rows.
>
> As a follow up experiment, we also evaluate the similarity of queries within a given gamma window, as similar queries transitively implies that rows of the attention matrix will have a similar distribution of attention scores, and therefore may be re-used in neighboring rows. We find the following correlations which tracks closely to that of Figure 6B.
>
> | $\gamma$ | 2 | 4 | 8 | 16 | 32 | 64 |
> |---|---|---|---|---|---|---|
> | cos(q, q') Avg. | 0.9584$\pm$0.0082 | 0.9095$\pm$0.0152 | 0.8592$\pm$0.0224 | 0.8119$\pm$0.0283 | 0.7768$\pm$0.0305 | 0.7488$\pm$0.0315 |
>
> ---
>
> > 2.1 Interpolation techniques might yield more accurate approximations.
>
> We tried this experiment by linearly interpolating between neighboring rows of delta terms instead of just plainly repeating them. We found that performance was largely unchanged by this fact. Additionally, linearly interpolating between delta terms introduces a theoretical concern of violating causality, as after the interpolation output token $i + \gamma$ has influenced the prediction of output token $i$, which is why we decided on repeating the delta correction in our original submission.
>
> The p-values are the result of a one sided paired permutation test (as suggested by reviewer 1VjR) to show whether there is a significant improvement added by the delta interpolation (p<0.05 is considered significant and therefore, there is no significant improvement seen by interpolation)
>
> |  | 131K | 65K | 32K | 16K | 8K | 4K | Avg. |
> |---|---|---|---|---|---|---|---|
> | Str.LLM + Delta | 64.40 | 75.22 | 81.27 | 88.66 | 92.25 | 96.54 | 83.06 |
> | Str.LLM + Delta (smooth) | 65.15 | 75.65 | 81.26 | 88.26 | 92.34 | 96.66 | 83.22 |
> | p-value (all non-qa tasks) | 0.1708 | 0.5185 | 0.3818 | 0.4279 | 0.5782 | 0.0001 | - |
> | p-value (all qa tasks) | 0.766 | 0.2455 | 0.8698 | 1 | 0.5043 | 1 | - |
>
> ---
>
> > 2.2 It's unclear how performance scales with the parameter (i.e., the fraction of queries used for full attention). At what value does the method match the performance of full attention?
>
> In our submission we have include an ablation in Figure 6a which shows how performance scales as a function of gamma.
>
> ---
>
> > Q.1 To what extent is the matrix query-dependent versus being a property of the model itself?
>
> If we understand the question correctly, you are asking if the direction of the delta correction is dependent on the query, or if the direction may be a fixed position learned within the model itself. To test this, we performed a variant of the experiment we did in Figure 6B. We took an input from RULER NIAH MultiKey3 and also an input from the QA2 task which is a more NLP oriented. We calculated the delta correction and averaged the cosine similarity within 6 gamma windows over all layers of Llama3.1-8b-instruct. The top row shows NIAH/QA2 cosine similarity average, where the cosine similarity was calculated within the gamma window of the current input (Self Avg. Same as Figure 6B but including a second task in the average). We also calculated the cosine similarity between the delta correction of the two tasks $\text{cos}(NIAH, QA)$ (Cross Avg).
>
> If the delta correction were a stable artifact of the model itself, we would expect to see a similar relationship in cosine similarity, i.e. it should start high for a small gamma window and slowly decrease as the gamma window increases). However, we find that between tasks, the cosine similarity between delta corrections remains low and shows no relationship with an increasing gamma window. Therefore, we can conclude that the delta correction is query/input dependent and not a fixed artifact of the model.
>
> Thank you for suggesting this experiment, we will add it to the final version of our work.
>
> | $\gamma$ | 2 | 4 | 8 | 16 | 32 | 64 |
> |---|---:|---:|---:|---:|---|---:|
> | NIAH/QA Self Avg. | 0.938 | 0.91545 | 0.9043 | 0.89165 | 0.8855 | 0.8774 |
> | NIAH/QA Cross Avg. | 0.1435 | 0.1417 | 0.1448 | 0.1417 | 0.1417 | 0.1425 |
>
> ---
>
> Thank you for your time and effort in evaluating our work. We will incorporate these new results into the final version of our work. We have done our best to respond to your concerns within the rebuttal period and remain open to further discussion throughout the discussion period.

---

> > ### Comment · Reviewer_7eBZ · 2025-08-04
> > **Thanks for the response.**
> >
> > Thanks for the rebuttal. The authors have addressed my question. I'll keep my positive rating.

---

> ### Author Response · Authors · 2025-08-05
>
> Thank you for your time and effort in reviewing our work, if there are any remaining concerns we remain open to further discussions until the very end of the discussion period.

---

### Official Review · Reviewer_QxjE · 2025-07-03

**Clarity:** 2
**Significance:** 2
**Originality:** 2
**Rating:** 4
**Confidence:** 4

**Summary:**

This paper addresses a critical limitation in sparse attention methods: the performance degradation caused by distributional shift in attention outputs. The authors propose ∆ Attention, a post-processing correction method that computes the difference between sparse and full attention for a subset of queries and applies this correction across all positions. The method achieves performance improvements on various tasks while maintaining computational efficiency.

**Questions:**

See weaknesses

**Ethical Concerns:**

["NO or VERY MINOR ethics concerns only"]

**Final Justification:**

My problems have been addressed. Therefore, I increase the score to 5.

**Limitations:**

See weaknesses

**Quality:**

2

**Strengths And Weaknesses:**

**Strengths:**
1. The proposed ∆ Attention is practical for improving accuracy with minimal computational overhead, which can be applied to various existing sparse attention techniques.
2. Excellent motivation by identifying the root cause of sparse attention degradation: distributional shift between dense and sparse attention
3. Evaluation across multiple benchmarks, including RULER, Infinite-Bench, and PG19

**Weaknesses:**
1. The paper fails to provide convincing theoretical evidence for why computing full attention for only sparse queries (every γ-th position) is sufficient to correct the distributional shift across all positions. The core assumption that (A∆V)i ≈ (A∆V)i+ν is supported merely by empirical cosine similarity measurements (Figure 6b), lacking rigorous theoretical grounding. This fundamental gap undermines our understanding of when and why the method works
2. The experimental evaluation suffers from two critical issues: (1): When comparing with KV cache eviction methods (e.g., StreamingLLM), the authors fail to explicitly acknowledge that ∆ Attention requires storing complete Key-Value pairs, dramatically increasing memory requirements from ~2K tokens to the full sequence length. This makes the comparison fundamentally unfair, as the methods operate under entirely different memory budgets. (2): For advanced sparse attention techniques with dynamic sparsity (HiP, MInference), the improvements are marginal (4-8%), raising questions about the method's cost-effectiveness when applied to already sophisticated approaches. The paper lacks an analysis of why the method shows such stark performance differences across baseline categories.
3. Some previous work has already worked on fixing the distribution shift for different kinds of sparse attention at test time, including:

      [1]. Dong, Harry, et al. "Get more with less: Synthesizing recurrence with kv cache compression for efficient llm inference." arXiv preprint arXiv:2402.09398 (2024).

      [2]. Yao, Jiayi, et al. "CacheBlend: Fast large language model serving for RAG with cached knowledge fusion." Proceedings of the Twentieth European Conference on Computer Systems. 2025.


      [3]. Sun Y, Ye T, Dong L, et al. Rectified Sparse Attention[J]. arXiv preprint arXiv:2506.04108, 2025.

      [4]. Yang X, Chen T, Chen B. APE: Faster and Longer Context-Augmented Generation via Adaptive Parallel Encoding[J]. arXiv preprint arXiv:2502.05431, 2025.

These methods will further refine the distribution at test time. However, the authors do not discuss about these related work.

---

> ### Author Rebuttal · Authors · 2025-07-30
>
> Thank you for your time and care in reviewing our work and noting the positive aspects of our method such as practicality, minimal overhead, easy integration, motivation, and evaluation. We will address you individual concerns in turn below.
>
> ---
>
> > 2.1 When comparing to Streaming LLM, the authors fail to acknowledge that delta attention requires storing all KV pairs, which increases the storage burden.
>
> We think this is a misunderstanding of our experimental setup. In our experiments, all models are allowed to see the full KV cache during decoding (including Streaming LLM). In this way, Streaming LLM is, in effect, similar to [2, 4, 5] (Cacheblend, APE, Star Attention), where the attention pattern is sparse during the prefill phase and dense during the decoding phase. We would like to highlight that even though Streaming LLM has access to all KV tokens during decode, it still mismatches queries and keys due to the distributional shift which our method is able to fix. We will update Lines 213-215 of our text to make this point clearer.
>
> ---
>
> > 2.2 For (HiP, MInference) improvements marginal (4-8%), raising questions about cost-effectiveness when applied to already sophisticated approaches. The paper lacks an analysis of why the method shows such stark performance differences across baseline categories
>
> We think this is quite an intuitive pattern. Naive sparse models like Streaming LLM have the most room for improvement because it shows the worst task performance. MInference and HiP are more sophisticated methods of selecting top-k attention which is to say that they have more precise approximation to the full attention matrix and therefore less room for improvement.
>
> We would like to note, however, that even though HiP and MInference are sophisticated methods, they still benefit (~4-8%) from applying our simple delta correction. We maintain that a 4-8% improvement on recent SOTA sparse attention methods is not trivial.
>
> ---
>
> > 3. Some previous work [1-4] has already worked on fixing the distribution shift for different kinds of sparse attention at test time
>
> **We would like to highlight the fact that [3] is not a previous work, as it was only published on arxiv on 6/4/2025 which is nearly a month after the NeurIPS paper deadline of 5/15/2025.** Therefore, it would have been impossible for us to know about this work at the time of submission. However, we will discuss any possible connection to the cited references below:
>
> Regarding [1]: The method titled LESS requires training a low rank cache compressor, while ours is a training-free modification. LESS also only experiments with a maximum of 4K context length which is well below the minimum length we tested (Our experiments go up to 384K). This work briefly mentions differences in attention distributions between dense and sparse attention. However they make no mention of the critical insight of our work, namely that a dense decode fails to properly align with the tokens resulting from a sparse prefill due to the distributional shift of the keys that is induced by the sparse prefill.
>
> Regarding [2]: Cacheblend proposes using one dense attention layer to identify important tokens, and then selectively recomputing these tokens in later layers to add missing parts of the sparse attention to cached KV pairs. Cacheblend proposed this as a way to augment and consolidate independently processed chunks of a RAG pipeline. In practice, however, this would effectively be similar to a "smart" sparse prefill method like HiP or MInference which fills in some of the missing tokens in the attention matrix which are outside of the local window context.
>
> We ran the officially published code for Cacheblend to compare with our method (results are posted below)
>
> Regarding [3]: **It would have been impossible for us to cite this work, as it was not even on arxiv at the time of submission.** However, we note that this work only considers dense prefills and a sparse decoding procedure. Their sparse decoding procedure is similar to what we call “recompute” in Table 4 and Figures 8,9,13-15. We showed that this “recomptue” method is insufficient to mitigate the distributional shift in the outputs.
>
> Regarding [4]: APE is in practice similar to [5] which we cited in our paper. The only difference we notice is that APE proposes temperature scaling and rescaling the attention post-hoc in order to correct any error introduced. However, APE misses the crucial insight of our work, namely that sparse and dense attention result in completely different token distributions which means that there is a problem of query-key matching during decoding. APE only considers query and key geometry as a function of a) position and b) input. They deduce that because they key states of the first few keys (sink tokens) are relatively stable, then the geometry of all other keys are also stable.
>
> Our crucial insight is that different attention operations themselves induce different distributions of query-key geometries and therefore when moving from sparse to dense during decoding (like APE does) there will be a mismatch between queries and keys. This is precisely what our delta correction fixes.
>
> We have run experiments against the officially published codebases of [2,4] on the RULER benchmark and find that our method results in better performance due to the delta correction.
>
> |  | 131K | 65K | 32K | 16K | 8K | 4K | Avg. |
> |---|---|---|---|---|---|---|---|
> | Str.LLM | 27.45 | 18.59 | 30.25 | 38.13 | 60.53 | 90.52 | 44.25 |
> | Cachblend [2] | 0.00 | 0.21 | 0.31 | 1.49 | 24.42 | 96.27 | 20.45 |
> | APE [4] | 26.76 | 43.03 | 53.13 | 67.50 | 77.25 | 93.76 | 60.24 |
> | Str.LLM + Delta | **64.40** | **75.22** | **81.27** | **88.66** | **92.25** | **96.54** | **83.06** |
>
> ---
>
> >  1. lacking theoretical grounding.
>
> In our submission, we were able to show that the difference between the full attention and the sparse attention output is an approximation to the missing attention region (of the full attention matrix). Figure 11 in the appendix shows that in practice, the approximation tends to be lower than the derived bound in Lemma 1. However, the applicability of the delta correction to subsequent rows of the output relies on local similarities of the queries within a given window (If queries are locally similar, then they will result in a similar row of the attention matrix and therefore a similar delta correction). Note that the queries receive the same ROPE positional adjustment as the keys, and therefore
>
> $\langle q_i', q_j' \rangle
> = \langle R(\theta_i)q_i, R(\theta_j)q_j \rangle
> = \langle q_i, R(\theta_j-\theta_i)q_j \rangle$
>
> So the queries are biased by ROPE to likely have a higher similarity within a local region. We measure the query similarity from inputs related to both NIAH and QA (natural language) tasks, and observe that across all layers, queries within a given $\gamma$-window show a reliable relationship of cosine similarities.
>
> | $\gamma$ | 2 | 4 | 8 | 16 | 32 | 64 |
> |---|---|---|---|---|---|---|
> | cos(q, q') Avg. | 0.9584$\pm$0.0082 | 0.9095$\pm$0.0152 | 0.8592$\pm$0.0224 | 0.8119$\pm$0.0283 | 0.7768$\pm$0.0305 | 0.7488$\pm$0.0315 |
>
> Also note that we tested our method in conjunction with Llama 4, which uses no ROPE embeddings on the dense attention layers, and we still show an improvement in accuracy on Infinitebench (Table 3 in our submission).
>
> ---
>
> ### References
>
> [5] Acharya, S., Jia, F., & Ginsburg, B. (2024). Star attention: Efficient llm inference over long sequences. *arXiv preprint arXiv:2411.17116*.
>
> ---
>
> Thank you for your time and effort in reviewing our work. We sincerely hope we have addressed the concerns you raised. We plan to add these additional results to the final version of our paper. We are available to answer any remaining concerns throughout the discussion period.

---

> > ### Comment · Reviewer_QxjE · 2025-07-31
> > **Response from Reviewer QxjE**
> >
> > Thank you very much for your rebuttal.
> >
> > First, I understand that you only use StreamingLLM for encoding and enable it to see all past KV states during decoding. However, this differs from the original intention of StreamingLLM, which focuses on memory-efficient and infinite-length generation. Therefore, I will define this as a weak baseline, as we all know it should have a huge problem in retrieval-related tasks. It is not reasonable to include this baseline when computing averaging improvements. We can still compare it separately. (I agree that improving 4-8% on these tasks is also a non-trivial improvement in comparison with other baselines.
> >
> > Second, I apologize for not noticing the publication date of [3]. For other works, I appreciate the author's comparison with Cacheblend and APE, which demonstrates a better ability to encode long context. Therefore, I think another possible experiment is to show that Star Attention/APE can be further improved with the delta rule. It will improve the quality of the paper.
> >
> > For comparison with LESS, I agree that the work focuses on sparse encoding and dense decoding, whereas LESS focuses on correcting sparse encoding and decoding with a low-rank cache. However, since there are not many methods available for correcting the difference between sparse and dense attention, I think a comparison is not necessary; however, it is reasonable to mention this in the related work.
> >
> > Moreover, I was wondering why we consider sparse encoding + dense decoding instead of sparse encoding + decoding. This scenario seems to be a little narrow, which only designed for long-context retrieval tasks.
> >
> > Thank you very much!

---

> ### Author Response · Authors · 2025-08-01
> **Star Attention Results (Part I)**
>
> Thank you for your prompt response to our rebuttal, we would like to provide more evidence of the novelty and significance of our contribution.
>
> > Therefore, I will define this as a weak baseline, as we all know it should have a huge problem in retrieval-related tasks. It is not reasonable to include this baseline when computing averaging improvements. We can still compare it separately.
>
> Streaming LLM with a dense decode has the nearly the same attention pattern as Star Attention and APE, therefore, we do not see why it should be considered a weak baseline. Furthermore, we disagree that “we all know it should have a problem in retrieval based tasks.” This was the precise question which motivated our work.
>
> To provide further evidence that it is not obvious, we direct your attention to Figure 4 and the conclusion of the Star Attention paper.
>
> >> Star Attention (Figure 4 Caption): All runs use a block and anchor block size set to one-quarter of the total sequence length
>
> >> Star Attention (Conclusion): The role and optimal size of anchor blocks relative to context blocks require further exploration. Additionally, while Star Attention performs effectively with block sizes set to one-quarter of the sequence length, accuracy degrades when using smaller blocks on longer sequences. Future work will focus on refining the anchor block mechanism.
>
> If the reason for requiring 1/4 of the context length were obvious the authors would not have suggested that the block sizing effect on accuracy needs further investigation.
>
> In fact, we started our investigation with the following question:
>
> ```
> If a sliding window (or block chunk) is large enough to fit relevant chunks of context, and all chunks of context are conditionally independent, then why should there be a problem retrieving the relevant context later when it can see all of the keys during decoding?
> ```
>
> We do not think it is sufficient to state this is obvious without precisely identifying why there should be any problem retrieving context which was fully contained in the sliding window. Our work precisely describes and identifies the mechanism which causes this to happen, and offers an effective solution which can be applied. Both the description of the cause and the solution have not been properly identified in prior work.
>
>
> > I think another possible experiment is to show that Star Attention/APE can be further improved with the delta rule. It will improve the quality of the paper.
>
> Please see Figure 2 in the Star Attention paper. This is almost identical to the attention pattern of Streaming LLM with a dense decode. This is why we chose to use Streaming LLM with dense decode as a baseline. To further illustrate that this is the case, we ran an experiment with a block mask **identical** to that of Star Attention with a block size of 2K and achieved the following results.
>
> | Star Attention Mask | niah_1 | niah_2 | niah_3 | niah_mk_1 | niah_mk_2 | niah_mk_3 | niah_mv | niah_mq | vt | cwe | fwe | qa_1 | qa_2 | Avg. |
> |-|-|-|-|-|-|-|-|-|-|-|-|-|-|-|
> | 128K | 2 | 2 | 10 | 2 | 4 | 0 | 4 | 7.5 | 3.2 | 0 | 91.33 | 14 | 16 | 12.00 |
> | 64K | 8 | 2 | 0 | 6 | 0 | 0 | 7.5 | 7 | 46 | 0 | 90.67 | 10 | 16 | 14.86 |
> | 32K | 2 | 6 | 10 | 14 | 8 | 2 | 34 | 24.5 | 25.6 | 0.8 | 96.67 | 18 | 24 | 20.43 |
> | 16K | 14 | 12 | 12 | 10 | 14 | 28 | 41 | 37 | 30.4 | 77.8 | 91.33 | 20 | 24 | 31.66 |
> | 8K | 32 | 32 | 38 | 44 | 36 | 48 | 63 | 68.5 | 62 | 92.6 | 88.67 | 28 | 38 | 51.60 |
> | 4K | 68 | 70 | 80 | 76 | 78 | 60 | 75 | 88.5 | 81.2 | 99.4 | 96 | 94 | 56 | 78.62 |
>
> | Star Attention Mask + Delta | niah_1 | niah_2 | niah_3 | niah_mk_1 | niah_mk_2 | niah_mk_3 | niah_mv | niah_mq | vt | cwe | fwe | qa_1 | qa_2 | Avg. |
> |-|-|-|-|-|-|-|-|-|-|-|-|-|-|-|
> | 128K | 96 | 100 | 100 | 86 | 36 | 18 | 42 | 77.5 | 30.8 | 0 | 82.67 | 64 | 32 | **58.84** |
> | 64K | 92 | 98 | 98 | 92 | 72 | 62 | 57 | 91.5 | 46 | 1 | 86 | 72 | 44 | **70.12** |
> | 32K | 88 | 96 | 94 | 90 | 86 | 82 | 76 | 90.5 | 43.6 | 5.2 | 96.67 | 72 | 52 | **74.77** |
> | 16K | 92 | 100 | 94 | 96 | 98 | 92 | 71.5 | 83.5 | 68.8 | 61.8 | 91.33 | 70 | 56 | **82.69** |
> | 8K | 98 | 100 | 100 | 94 | 100 | 96 | 75 | 88 | 84.8 | 91.4 | 93.33 | 80 | 58 | **89.12** |
> | 4K | 100 | 98 | 94 | 96 | 100 | 98 | 86.5 | 97.5 | 93.2 | 98.8 | 94.67 | 90 | 66 | **93.28** |
>
> This result provides further evidence towards the fact that our delta correction can improve upon the block attention pattern of Star Attention **and** that Streaming LLM with dense decode should not be considered a weak baseline as it outperforms the block attention mask of Star Attention due to the added information which is missing in the block structure of the Star Attention mask.
>
> ---
>
> Continued...

---

> ### Author Response · Authors · 2025-08-01
> **Star Attention Results (Part II)**
>
> > For comparison with LESS, I agree that the work focuses on sparse encoding and dense decoding, whereas LESS focuses on correcting sparse encoding and decoding with a low-rank cache. However, since there are not many methods available for correcting the difference between sparse and dense attention, I think a comparison is not necessary; however, it is reasonable to mention this in the related work.
>
> We will mention this work and the other mentioned references [1-4] in the final version of our related work.
>
> ---
>
> > Moreover, I was wondering why we consider sparse encoding + dense decoding instead of sparse encoding + decoding.
>
> We agree that sparse decoding is an interesting next step. However the main purpose of our work was in identifying the cause of failure in the sparse prefill/dense decode setting and proposing a solution which is effective in mitigating the problem.
>
> > This scenario seems to be a little narrow, which only designed for long-context retrieval tasks.
>
> Our setup is not only designed for long context retrieval, as we also present results on InfiniteBench (Table 3) which includes Question Answering, Multiple Choice Questions, and Summarization, all of which are NLP-based and not retrieval tasks.
>
> —
>
> We thank you again for your review and responses, we hope this added evidence has demonstrated the strength and novelty of our analysis of the problem and our proposed solution. We remain open to further discussion until the end of the discussion period.

---

> > ### Comment · Reviewer_QxjE · 2025-08-05
> > **Response from Reviewer QxjE**
> >
> > Thank you very much for your further explanation!
> >
> > I agree that StreamingLLM w/ dense decoding is similar to Star Attention. So I agree that both baselines should be considered. However, I would appreciate it if the authors could make some claims about the separate improvements in both cases (static sparse attention and dynamic ones). This will be better for the audience to understand these improvements. However, I have to mention that Star Attention's design also focuses on reducing communication cost between hosts while APE focuses on RAG settings with/ context precomputation. Therefore, they have some specific improvements in addition to reduce the computation cost from quadratic to linear complexity.
> >
> > I think this method should be sufficient for long prefilling and short decoding. While I believe many long prefilling tasks require long decoding, I agree that this also serves as a mainstream category.
> >
> > Based on this information, I would like to increase my score to 4. (In case the openreview hides the modified score). Thank you very much for your time.

---

> ### Author Response · Authors · 2025-08-05
>
> Thank you for your continued discussion and evaluation of our comments. We agree that Star Attention and APE focus on contributions related to parallelization over hosts, and RAG precomputation which are orthogonal to our work which focuses on the distributional shift which can result from such attention patterns when switching to decoding.
>
> We will highlight this distinction when mentioning these methods in our final paper, and we hope that our contribution will inspire future works to devise methods of mitigating the distributional shift in distributed or precomputed chunked prefills.
>
> Thank you for your time and effort in reviewing our work.

---

### Author Response · Authors · 2025-08-09
**Discussion Period Summary**

Dear Reviewers,

We are pleased that after discussion, everyone seems to agrees on acceptance. We would like to summarize the new results from the discussion period below:

---

# New Results

**Baseline Comparisons:**

- Added comparison to APE, Star Attention, Star Attention + Delta (Ours), and Cacheblend.
- Each of these works shows a similar test-time attention pattern as ours (sparse prefill, dense decode) but none of them have accurately identified and mitigated the distributional shift between sparse prefill and dense decoding which harms the overall performance of the model.
- Our experiments show that Delta Attention is simple, easy to integrate with existing methods, and provides an effective correction to this distributional shift in tokens caused by sparse prefills, resulting in better performance.

---

**We have added experiments utilizing three types of interpolation between deltas:**

- Linear interpolation between one delta and the subsequent delta (which has theoretical concerns with breaking causality)
- Left-side only interpolation by EMA
- Left-side only interpolation/imputation by $\alpha,\beta,\gamma$-filter

Our original method outperformed all types of interpolation, although we agree with the intuition that approaches which address the discrete jumps in the delta correction are a likely avenue for further research efforts.

---

**We have provided an experiment with reasoning models:**

- This was done on the RepoQA dataset with Qwen3 235B A22B
- The experiment shows that our model can be integrated with reasoning tasks and still improve performance

---

**Ablations and Misc. Experiments. We have:**

- Provided a further ablation which shows that the delta correction is query dependent and not a fixed direction within the model.
- Provided paired permutation tests show our models improvement is statistically significant.

---

# Positive Comments from Reviewers

- **(QxjE)** Excellent motivation by identifying the root cause of sparse attention degradation: distributional shift between dense and sparse attention
  - practical for improving accuracy with minimal computational overhead, which can be applied to various existing sparse attention techniques.

- **(7eBZ)** The idea of using a small number of full attention computations to correct or enhance sparse attention scores is novel and interesting
  - The experimental results effectively demonstrate the benefits of the approach.

- **(1VjR)** Interesting new method which improves quality for sparse attention models.
  - Extensive empirical results; clear presentation

- **(wJZr)** The analysis about the distribution shift of attention scores shown in fig 3 is insightful
  - performance gains seem significant; increased latency is minimum
  - the ablations about delta and recomputation are comprehensive
  - This method is extensively evaluated on several long context tasks, achieving superior performances than existing methods while keeping sparsity.
  - As reported, this method could be seamlessly integrated with existing sparse methods.

---

We thank the reviewers for their work in evaluating our submission. We will add all new experiments to the revised version of our work. We sincerely hope we have addressed all concerns during the discussion period.

Thank you.

---

### Decision · Program_Chairs · 2025-09-17

**Decision:**

Accept (poster)

**Comment:**

This paper proposes a new sparse attention method to reduce computational complexity of self-attention during the model inference. The proposed method identifies the distribution shift between the full attention and sparse attention as a source of performance degradation, and proposes a delta method to correct this distribution gap. Empirical evaluation on several long context tasks indicate superior performances than baselines while keeping sparsity. Overall, reviewers found this paper well written and with a clear motivation. The paper sheds light on the distribution shift between the full attention and sparse attention which is insightful and useful. The empirical analysis is solid and includes ablation experiments. Some of the weaknesses have been well addressed during the rebuttal stage. I recommend acceptance.